# MOIRAI-MOE: EMPOWERING TIME SERIES FOUNDATION MODELS WITH SPARSE MIXTURE OF EXPERTS

## ABSTRACT

Time series foundation models have demonstrated impressive performance as zero-shot forecasters, i.e., they can tackle a wide variety of downstream forecasting tasks without explicit task-specific training. However, achieving effectively unified training on time series remains an open challenge. Existing approaches introduce some level of model specialization to account for the highly heterogeneous nature of time series data. For instance, MOIRAI pursues unified training by employing multiple input/output projection layers, each tailored to handle time series at a specific frequency. Similarly, TimesFM maintains a frequency embedding dictionary for this purpose. We identify two major drawbacks to this human-imposed frequency-level model specialization: (1) Frequency is not a reliable indicator of the underlying patterns in time series. For example, time series with different frequencies can display similar patterns, while those with the same frequency may exhibit varied patterns. (2) Non-stationarity is an inherent property of real-world time series, leading to varied distributions even within a short context window of a single time series. Frequency-level specialization is too coarse-grained to capture this level of diversity. To address these limitations, this paper introduces MOIRAI-MOE, using a single input/output projection layer while delegating the modeling of diverse time series patterns to the sparse mixture of experts (MoE) within Transformers. With these designs, MOIRAI-MOE reduces reliance on human-defined heuristics and enables automatic token-level specialization. Extensive experiments on 39 datasets demonstrate the superiority of MOIRAI-MOE over existing foundation models in both in-distribution and zero-shot scenarios. Furthermore, this study conducts comprehensive model analyses to explore the inner workings of time series MoE foundation models and provides valuable insights for future research.

## 1 INTRODUCTION

Foundation models have transformed several fields, such as natural language processing (Dubey et al., 2024) and computer vision (Kirillov et al., 2023), demonstrating impressive zero-shot performance. Inspired by these successes, time series forecasting is experiencing a similar shift (Liang et al., 2024). The traditional approach of developing separate models for each dataset is being replaced by the concept of universal forecasting (Woo et al., 2024), where a pretrained model can be applied across diverse downstream tasks in a zero-shot manner, regardless of variations in domain, frequency, dimensionality, context, or prediction length. This new paradigm significantly reduces the complexity of building numerous specialized models, paving the way for forecasting-as-a-service.

To excel in zero-shot forecasting, time series foundation models are pretrained on massive data from a variety of sources. However, unlike language and vision modalities which benefit from standardized input formats, time series data is inherently heterogeneous, posing significant challenges for *unified time series training*. Existing solutions such as TEMPO (Cao et al., 2024) and UniTime (Liu et al., 2024a) leverage language prompts to provide data identification information, thereby discerning the source of data and achieving model specialization at the dataset level. MOIRAI (Woo et al., 2024) goes a step further and proposes a more granular categorization based on a time series meta feature – frequency. Specifically, they design multiple input/output projection layers with each layer specialized to handle data corresponding to a specific frequency, thereby enabling frequency-level specialization. Similarly, TimesFM (Das et al., 2024) is also at this level of specialization, distinguishing the data by maintaining a frequency embedding mapping.

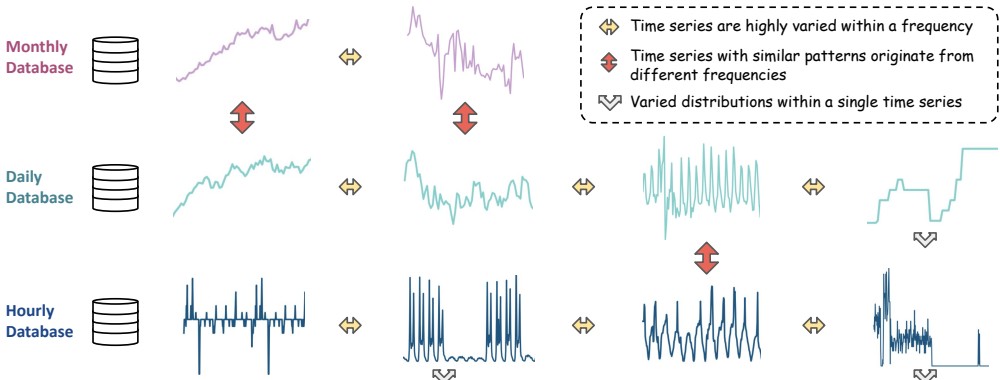

Figure 1: An illustration of the challenges arising from grouping time series by frequency and imposing frequency-level model specialization: the diversity of patterns within the same frequency group, the similarity of patterns across different frequencies, and the variability of distributions within a single time series. The examples presented are derived from **real time series** in the Monash benchmark (Godahewa et al., 2021).

Given the heterogeneity of time series, we acknowledge the value of model specialization; however, we argue that *human-imposed frequency-level specialization lacks generalizability and introduces several limitations*. (1) Frequency is not always a reliable indicator and might not effectively capture the true structure of time series data. As shown in Figure 1, time series with different frequencies can exhibit similar patterns, while those with the same frequency may display diverse and unrelated patterns. This human-imposed mismatch between frequency and pattern undermines the efficacy of model specialization, resulting in inferior performance. (2) Furthermore, real-world time series are inherently non-stationary (Liu et al., 2022), displaying varied distributions even within a short context window of a single time series. Clearly, frequency-level specialization is too coarse-grained to capture this level of diversity, underscoring the need for more fine-grained modeling approaches.

To address the aforementioned issues, this paper introduces **MOIRAI-MOE**, an innovative solution for effective time series unified training, inspired by recent developments of Sparse Mixture of Experts (MoE) Transformers (Lepikhin et al., 2021; Fedus et al., 2022; Dai et al., 2024). The core idea of MOIRAI-MOE is to utilize a single input/output projection layer while delegateing the modeling of diverse time series patterns to the sparse specialized experts in Transformer layers. With these designs, specialization of MOIRAI-MOE is achieved in a data-driven manner and operates at the token level. Moreover, this study investigates existing expert gating functions that generally use a randomly initialized linear layer for expert assignments (Shazeer et al., 2017; Jiang et al., 2024) and introduces a new function that leverages cluster centroids derived from a pretrained model to guide expert allocations.

We extensively evaluate MOIRAI-MOE using a total of 39 datasets in in-distribution and zero-shot forecasting scenarios. The results confirm the superiority of MOIRAI-MOE over state-of-the-art foundation models including TimesFM (Das et al., 2024), Chronos (Ansari et al., 2024), and MOIRAI (Woo et al., 2024). Additionally, we conduct comprehensive model analyses, as the first attempt, to explore the inner workings of time series MoE foundation models. It reveals that MOIRAI-MOE acquires the capability to achieve frequency-invariant representations and essentially performs progressive denoising throughout the model. Our contributions are summarized as follows:

- We propose MOIRAI-MOE, the first mixture-of-experts time series foundation model, achieving token-level model specialization in a data-driven manner. We introduce a new expert gating function for accurate expert assignments and improved performance.

- Extensive experiments on 39 datasets reveal that MOIRAI-MOE delivers up to 17% performance improvements over MOIRAI at the same level of model size, and outperforms other time series foundation models with up to $65\times$ fewer activated parameters.

- We conduct thorough model analyses to deepen understanding of the inner workings of time series MoE foundation models and summarize valuable insights for future research.

## 2 RELATED WORK

**Foundation Models for Time Series Forecasting** Time series foundation models serve as versatile zero-shot forecasting tools. A key challenge in training these models is accommodating the high diversity of time series data, underscoring the possible need for designing specialization modules. Current approaches like TEMPO (Cao et al., 2024) and UniTime (Liu et al., 2024a) utilize language-based prompts to identify data sources, facilitating model specialization at the dataset level. MOIRAI (Woo et al., 2024) advances this by focusing on a time series meta feature – frequency. This method designs separate input/output projection layers for specific frequencies, allowing for frequency-specific specialization. Similarly, TimesFM (Das et al., 2024) operates at this level of specialization by incorporating a frequency embedding dictionary to differentiate data. Some methods, like Chronos (Ansari et al., 2024), Lag-LLaMA (Rasul et al., 2023), Moment (Goswami et al., 2024), and Timer (Liu et al., 2024c), do not incorporate any specialization modules. Instead, they utilize the same architecture for all time series data, which can potentially increase the learning complexity and demand a large number of parameters to memorize the diverse input patterns. In this work, we propose to achieve automatic token-level specialization by using sparse mixture of experts, where diverse time series tokens are processed by specialized experts, while similar tokens share parameter space, thereby reducing learning complexity.

**Sparse Mixture of Experts** Mixture of experts (MoE) has emerged as an effective method for significantly scaling up model capacity while minimizing computation overhead in Large Language Models (LLMs) (Fedus et al., 2022; Dai et al., 2024; Zhu et al., 2024). A common approach for integrating MoE into Transformers involves replacing Feed-Forward Networks (FFNs) with MoE layers. An MoE layer consists of multiple expert networks and a gating function, where each expert shares the same structure as a standard FFN. The gating function is responsible for producing a gating vector that indicates the expert assignment. The assignment is usually sparse to maintain computational efficiency in the MoE layer, meaning that each token is generally processed by only one (Fedus et al., 2022) or two (Rajbhandari et al., 2022; Jiang et al., 2024) experts. In time series forecasting, several studies employ the concept of mixture of experts (Zeevi et al., 1996; Yuksel et al., 2012; Ni et al., 2024). In their contexts, the term experts typically refers to linear-centric models, such as autoregressive linear models and DLinear (Zeng et al., 2023). However, these methods are trained on specific datasets, limiting their ability to generalize and function as foundation models.

## 3 METHODOLOGY

In this section, we present MOIRAI-MOE, a mixture-of-experts time series foundation model built upon MOIRAI (Woo et al., 2024). Figure 2 presents a comparison. While MOIRAI-MOE inherits many of the strengths of MOIRAI, its major enhancement lies in: rather than using multi heuristic-defined input/output projection layers to model time series with different frequencies, MOIRAI-MOE utilizes a single input/output projection layer while delegating the task of capturing diverse time series patterns to the sparse mixture of experts in the Transformer. In addition, MOIRAI-MOE proposes a novel gating function that leverages knowledge from a pretrained model, and adopts a decoder-only training objective to improve training efficiency by enabling parallel learning of various context lengths in a single model update. We describe each model component in the following parts.

### 3.1 TIME SERIES TOKEN CONSTRUCTION

Patching techniques, first introduced in PatchTST (Nie et al., 2023), have become a prevalent method in many state-of-the-art time series models (Das et al., 2024; Liu et al., 2024a; Woo et al., 2024). By aggregating adjacent time series data into patches, this technique effectively captures local semantic information and significantly reduces computational overhead when processing long inputs. Given a time series with length $S$, we segment it into non-overlapping patches of size $P$, resulting in a sequence of patches $\boldsymbol{x} \in \mathbb{R}^{N \times P}$, where $N = \lceil \frac{S}{P} \rceil$.

We then normalize the patches to mitigate distribution shift issues (Liu et al., 2022; Wu et al., 2023). In a decoder-only (autoregressive) model, where each patch predicts its succeeding patch, applying a causal normalizer to each patch is the most effective way to achieve accurate normalization. However, this approach generates $N$ subsequences with different lengths, diminishing the parallel training

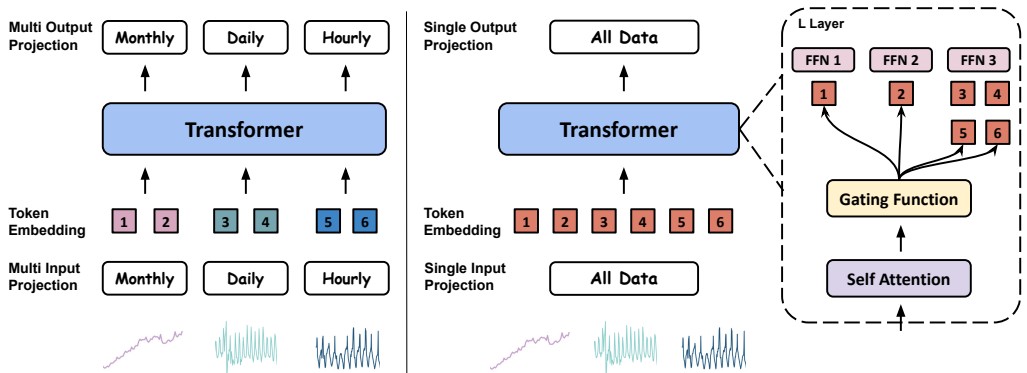

Figure 2: Comparison of MOIRAI (left) and MOIRAI-MOE (right).

that decoder-only models typically offer. To address this, we introduce the masking ratio $r$ as a hyperparameter, which specifies the portion of the entire sequence used exclusively for robust normalizer calculation, without contributing to the prediction loss. Finally, we forward the patches through a single projection layer to generate time series tokens $\boldsymbol{x} \in \mathbb{R}^{N \times D}$, where $D$ is the dimension of Transformers. We pass on the capability of learning diverse time series patterns to the vast number of parameters in Transformers. This projection layer is implemented as a residual multi-layer perceptron to enhance representation capacity (Das et al., 2023).

### 3.2 MIXTURE OF EXPERTS FOR TRANSFORMERS

A decoder-only Transformer (Dubey et al., 2024) is constructed by stacking $L$ layers of Transformer blocks. The block at the $l$-th layer is represented as follows:

$$\tilde{\boldsymbol{x}}^l = \text{CSA}(\text{LN}(\boldsymbol{x}^l)) + \boldsymbol{x}^l \tag{1}$$

$$\boldsymbol{x}^{l+1} = \text{FFN}(\text{LN}(\tilde{\boldsymbol{x}}^l)) + \tilde{\boldsymbol{x}}^l \tag{2}$$

where $\tilde{\boldsymbol{x}}^l \in \mathbb{R}^{N \times D}$ are the hidden states of all tokens after the attention module of the $l$-th layer and $\boldsymbol{x}^l = \boldsymbol{x}^{l+1} \in \mathbb{R}^{N \times D}$ are the input and output hidden states of the $l$-th layer. CSA, FFN, and LN denote a causal self-attention module, a feed-forward network, and the layer normalization, respectively. Following MOIRAI (Woo et al., 2024), MOIRAI-MOE captures multivariate correlations by flattening all variates into a sequence. During causal attention, each token is allowed to attend to its preceding tokens, as well as preceding tokens from other variates.

Next, we establish the mixture of experts by replacing each FFN with a MoE layer, which is composed of $M$ expert networks $\{E_1, \ldots, E_M\}$ and a gating function $G$. Only a subset of experts is activated for each token, allowing experts to specialize in distinct patterns of time series data and ensuring computational efficiency. The output of the MoE layer is computed as:

$$\sum_{i=1}^{M} G(\tilde{\boldsymbol{x}}^l)_i \cdot E_i(\tilde{\boldsymbol{x}}^l) \tag{3}$$

where $E_i(\tilde{\boldsymbol{x}}^l)$ is the output of the $i$-th expert network, and $G(\tilde{\boldsymbol{x}}^l)_i$ is the $i$-th token-to-expert affinity score generated by the gating function. Following Lepikhin et al. (2021); Rajbhandari et al. (2022); Jiang et al. (2024), we set the number of activated experts to $K = 2$.

#### 3.2.1 GATING FUNCTION

**Linear Projection as Gating Function.** A popular and effective gating function takes the softmax over the TopK logits of a linear projection parameterized by $\boldsymbol{W}_g \in \mathbb{R}^{D \times M}$ (Shazeer et al., 2017; Jiang et al., 2024; Dai et al., 2024):

$$G(\tilde{\boldsymbol{x}}^l) = \text{Softmax}(\text{TopK}(\tilde{\boldsymbol{x}}^l \cdot \boldsymbol{W}_g)) \tag{4}$$

However, the sparse gating can result in a load balancing issue (Shazeer et al., 2017). To mitigate this, an auxiliary loss is typically introduced to encourage an even distribution of tokens across experts

(Lepikhin et al., 2021; Fedus et al., 2022; Jiang et al., 2024; Dai et al., 2024). Formally, the load balancing loss for a batch $\mathcal{B}$ containing $T$ tokens is defined as:

$$\mathcal{L}_{\text{load}} = \sum_{i=1}^{M} \mathcal{D}_i \mathcal{P}_i, \text{ where } \mathcal{D}_i = \frac{1}{T} \sum_{t=1}^{T} \mathbb{1}\{\text{Token t selects Expert i}\}, \mathcal{P}_i = \frac{1}{T} \sum_{t=1}^{T} G(\tilde{\boldsymbol{x}}^l)_i \quad (5)$$

where $\mathbb{1}$ is the indicator function, $\mathcal{D}_i$ denotes the fraction of tokens routed to expert $i$, and $\mathcal{P}_i$ indicates the proportion of the gating probability allocated to expert $i$. The loss $\mathcal{L}_{\text{load}}$ is applied to each Transformer layer $l$. It is then aggregated by computing the mean across all layers and added to the prediction loss $\mathcal{L}_{\text{pred}}$ with a weight of 0.01 (Jiang et al., 2024; Dai et al., 2024).

**Token Clusters as Gating Function.** In this work, we propose a new gating mechanism that leverages cluster centroids derived from the token representations of a pretrained model to guide expert allocations. The intuition behind this approach is that clusters of pretrained token embeddings more closely reflect the real distribution of the data, leading to more effective expert specialization compared to a randomly initialized linear projection layer. Specifically, we first pretrain a MOIRAI model using single-patch input/output projection layers to mitigate the human-imposed frequency biases in MOIRAI. We then perform inference using our pretraining data. For a batch $\mathcal{B}$ containing $T$ tokens, we extract the attention outputs $\tilde{\boldsymbol{x}}^l \in \mathbb{R}^{T \times D}$ at each layer and perform mini-batch k-means clustering on them to continuously learn clusters at each layer. The number of clusters is set to match the total number of experts. During MoE training, for each layer, each token computes the Euclidean distance to learned cluster centroids $\boldsymbol{C} \in \mathbb{R}^{M \times D}$, and these distances serve as token-to-expert affinity scores for expert assignments:

$$G(\tilde{\boldsymbol{x}}^l) = \text{Softmax}(\text{TopK}(\text{Euclidean}(\tilde{\boldsymbol{x}}^l, \boldsymbol{C}))) \quad (6)$$

## 3.3 Training Objective

Let $\boldsymbol{x}_{t-l+1:t} = \{\boldsymbol{x}_{t-l+1}, \ldots, \boldsymbol{x}_t\}$ denote the context window of length $l$ for a token at position $t$. In this study, to facilitate both point and probabilistic forecasting, our goal is formulated as forecasting the predictive distribution of the next token $p(\boldsymbol{x}_{t+1}|\phi)$ by predicting the mixture distribution parameters $\hat{\phi}$ (Woo et al., 2024). These parameters are derived from the output tokens of the Transformer, followed by a single output projection layer. The following negative log-likelihood is minimized during training:

$$\mathcal{L}_{\text{pred}} = -\log p(\boldsymbol{x}_{t+1}|\hat{\phi}), \ \hat{\phi} = f_{\boldsymbol{\theta}}(\boldsymbol{x}_{t-l+1:t}) \quad (7)$$

## 4 Experiments

### 4.1 Moirai-MoE Setup

To ensure a fair comparison with MOIRAI in terms of activated parameters, we configure the number of activated experts as $K = 2$ for MOIRAI-MOE, resulting in 11M/86M activated parameters per token for MOIRAI-MOE$_\text{S}$/MOIRAI-MOE$_\text{B}$, closely matching the dense model MOIRAI$_\text{S}$/MOIRAI$_\text{B}$ that contains 14M/91M activated parameters. The total number of experts $M$ is set to 32, yielding total parameter sizes of 117M for MOIRAI-MOE$_\text{S}$ and 935M for MOIRAI-MOE$_\text{B}$. MOIRAI-MOE$_\text{L}$ is not presented due to the significant requirements of computational resources. The specific configurations are outlined in Table 1.

Table 1: Model configurations of MOIRAI and MOIRAI-MOE.

| Model | Layers | $d_{\text{model}}$ | $d_{\text{ff}}$ | Activated Params | Total Params | Activated Experts | Total Experts |
|---|---|---|---|---|---|---|---|
| MOIRAI$_\text{S}$ | 6 | 384 | 1,024 | 14M | 14M | – | – |
| MOIRAI$_\text{B}$ | 12 | 768 | 2,048 | 91M | 91M | – | – |
| MOIRAI$_\text{L}$ | 24 | 1,024 | 2,736 | 310M | 310M | – | – |
| MOIRAI-MOE$_\text{S}$ | 6 | 384 | 512 | 11M | 117M | 2 | 32 |
| MOIRAI-MOE$_\text{B}$ | 12 | 768 | 1,024 | 86M | 935M | 2 | 32 |

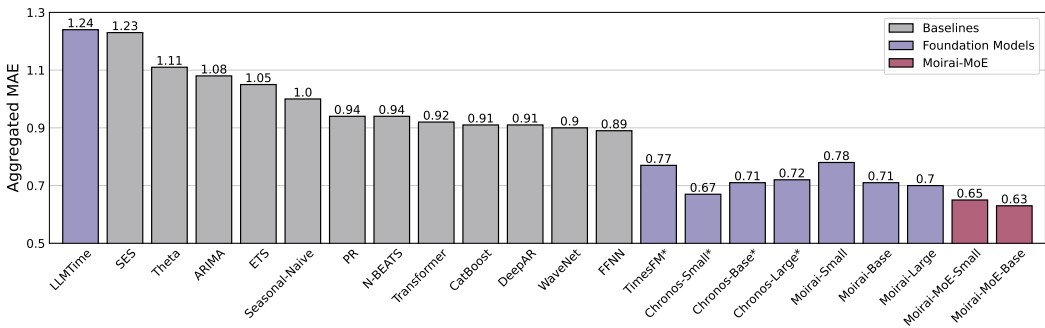

Figure 3: In-distribution forecasting evaluation using **29** datasets from Monash (Godahewa et al., 2021). Asterisks (*) indicate that the methods incorporated the datasets here in their pretraining corpora. The aggregate MAE is reported. For each dataset, the MAE is normalized by the MAE of the seasonal naive forecast, and the results are combined using the geometric mean. Dataset details and full results are provided in Appendix A.1. Context lengths of methods are in Table 9.

Table 2: Zero-shot performance of probabilistic and point forecasting. Asterisks (*) indicate the non-zero-shot datasets. The Avg column is normalized by seasonal naive, followed by geometric mean. Two Avg values are shown: one that averages all data, and another (non-leak) excludes Electricity and Solar. Best average results are highlighted in **red**, and second best results are in **blue**. Power: Turkey Power. Traffic: Istanbul Traffic. Weather: Jena Weather. BizITObs: BizITObs-L2C. Dataset details and context lengths of methods are in Table 7 and 9.

| Method | Metric | Electricity | Solar | Power | ETT1 | ETT2 | Traffic | MDENSE | Walmart | Weather | BizITObs | Avg (all) | Avg (non-leak) |
|---|---|---|---|---|---|---|---|---|---|---|---|---|---|
| Seasonal Naive | CRPS | 0.070 | 0.512 | 0.085 | 0.515 | 0.205 | 0.257 | 0.294 | 0.151 | 0.068 | 0.262 | 1.000 | 1.000 |
|  | MASE | 0.881 | 1.203 | 0.906 | 1.778 | 1.390 | 1.137 | 1.669 | 1.236 | 0.782 | 0.986 | 1.000 | 1.000 |
| TiDE | CRPS | 0.048 | 0.420 | 0.046 | 1.056 | 0.130 | 0.110 | 0.091 | 0.077 | 0.054 | 0.124 | 0.631 | 0.604 |
|  | MASE | 0.706 | 1.265 | 0.904 | 6.898 | 2.189 | 0.618 | 0.911 | 0.814 | 0.832 | 0.450 | 0.931 | 0.934 |
| PatchTST | CRPS | 0.052 | 0.518 | 0.054 | 0.304 | 0.131 | 0.112 | 0.070 | 0.082 | 0.059 | 0.074 | 0.549 | 0.490 |
|  | MASE | 0.753 | 1.607 | 1.234 | 1.680 | 2.168 | 0.653 | 0.732 | 0.867 | 0.844 | 0.266 | 0.808 | 0.753 |
| iTransformer | CRPS | 0.057 | 0.443 | 0.056 | 0.344 | 0.129 | 0.105 | 0.072 | 0.070 | 0.053 | 0.077 | 0.540 | 0.483 |
|  | MASE | 0.875 | 1.342 | 1.076 | 2.393 | 1.841 | 0.581 | 0.727 | 0.761 | 0.623 | 0.271 | 0.767 | 0.708 |
| MoLE-DLinear | CRPS | 0.083 | 0.535 | 0.072 | 0.344 | 0.188 | 0.237 | 0.108 | 0.137 | 0.079 | 0.095 | 0.780 | 0.714 |
|  | MASE | 0.984 | 1.257 | 1.325 | 1.606 | 3.194 | 1.016 | 0.914 | 1.115 | 0.925 | 0.282 | 0.938 | 0.906 |
| TimesFM | CRPS | 0.045* | 0.456 | 0.037 | 0.280 | 0.113 | 0.131 | 0.070 | 0.067 | 0.042 | 0.080 | **0.488** | **0.439** |
|  | MASE | 0.655* | 1.391 | 0.851 | 1.700 | 1.644 | 0.678 | 0.702 | 0.735 | 0.440 | 0.310 | 0.689 | 0.640 |
| TTM | CRPS | 0.075 | 0.534* | 0.059 | 0.417 | 0.122 | 0.210 | 0.150 | 0.192 | 0.055 | 0.102 | 0.758 | 0.697 |
|  | MASE | 0.802 | 1.255* | 0.898 | 1.934 | 1.547 | 0.901 | 1.195 | 1.477 | 0.506 | 0.308 | 0.831 | 0.798 |
| Timer | CRPS | 0.084 | 0.573 | 0.066 | 0.345 | 0.135 | 0.182 | 0.152 | 0.151 | 0.092 | 0.120 | 0.797 | 0.726 |
|  | MASE | 0.967 | 1.344 | 1.006 | 1.697 | 1.754 | 0.770 | 1.196 | 1.219 | 0.655 | 0.376 | 0.871 | 0.820 |
| Moment | CRPS | 0.354 | 1.332 | 0.151 | 0.401 | 0.277 | 0.612 | 0.157 | 0.154 | 0.105 | 0.313 | 1.502 | 1.205 |
|  | MASE | 3.167 | 3.139 | 2.244 | 2.243 | 4.100 | 2.617 | 1.277 | 1.245 | 1.053 | 0.913 | 1.691 | 1.457 |
| Chronos$_S$ | CRPS | 0.043* | 0.389* | 0.038 | 0.360 | 0.097 | 0.124 | 0.087 | 0.079 | 0.089 | 0.087 | 0.543 | 0.513 |
|  | MASE | 0.629* | 1.193* | 0.717 | 1.799 | 1.431 | 0.622 | 0.834 | 0.849 | 0.606 | 0.301 | 0.694 | 0.661 |
| Chronos$_B$ | CRPS | 0.041* | 0.341* | 0.039 | 0.387 | 0.092 | 0.109 | 0.075 | 0.080 | 0.058 | 0.084 | 0.499 | 0.471 |
|  | MASE | 0.617* | 1.002* | 0.722 | 1.898 | 1.265 | 0.553 | 0.712 | 0.849 | 0.583 | 0.301 | **0.656** | 0.631 |
| Chronos$_L$ | CRPS | 0.041* | 0.339* | 0.038 | 0.404 | 0.091 | 0.117 | 0.075 | 0.073 | 0.062 | 0.084 | 0.500 | 0.473 |
|  | MASE | 0.615* | 0.987* | 0.702 | 1.959 | 1.270 | 0.597 | 0.724 | 0.788 | 0.601 | 0.310 | 0.660 | 0.638 |
| MOIRAI$_S$ | CRPS | 0.072 | 0.471 | 0.048 | 0.275 | 0.101 | 0.173 | 0.084 | 0.103 | 0.049 | 0.081 | 0.578 | 0.507 |
|  | MASE | 0.981 | 1.465 | 0.948 | 1.701 | 1.417 | 0.990 | 0.836 | 1.048 | 0.521 | 0.301 | 0.798 | 0.726 |
| MOIRAI$_B$ | CRPS | 0.055 | 0.419 | 0.040 | 0.301 | 0.095 | 0.116 | 0.104 | 0.093 | 0.041 | 0.078 | 0.520 | 0.467 |
|  | MASE | 0.792 | 1.292 | 0.888 | 1.736 | 1.314 | 0.644 | 1.101 | 0.964 | 0.487 | 0.291 | 0.736 | 0.685 |
| MOIRAI$_L$ | CRPS | 0.050 | 0.406 | 0.036 | 0.286 | 0.094 | 0.112 | 0.095 | 0.098 | 0.051 | 0.079 | 0.514 | 0.467 |
|  | MASE | 0.751 | 1.237 | 0.870 | 1.750 | 1.436 | 0.631 | 0.957 | 1.007 | 0.515 | 0.285 | 0.729 | 0.685 |
| Time-MoE$_B$ | CRPS | 0.051* | 0.230* | 0.044 | 0.392 | 0.125 | 0.152 | 0.099 | 0.100 | 0.070 | 0.112 | 0.583 | 0.586 |
|  | MASE | 0.587* | 0.535* | 0.800 | 1.823 | 1.672 | 0.672 | 0.846 | 0.833 | 0.558 | 0.343 | 0.662 | 0.695 |
| Time-MoE$_L$ | CRPS | 0.051* | 0.294* | 0.045 | 0.386 | 0.131 | 0.172 | 0.090 | 0.097 | 0.058 | 0.111 | 0.589 | 0.576 |
|  | MASE | 0.581* | 0.689* | 0.790 | 1.773 | 1.878 | 0.762 | 0.759 | 0.817 | 0.524 | 0.337 | 0.678 | 0.695 |
| MOIRAI-MoE$_S$ | CRPS | 0.046 | 0.429 | 0.036 | 0.288 | 0.093 | 0.108 | 0.071 | 0.090 | 0.056 | 0.081 | 0.497 | **0.450** |
|  | MASE | 0.719 | 1.222 | 0.737 | 1.750 | 1.248 | 0.563 | 0.746 | 0.927 | 0.476 | 0.298 | 0.670 | **0.620** |
| MOIRAI-MoE$_B$ | CRPS | 0.041 | 0.382 | 0.034 | 0.296 | 0.091 | 0.100 | 0.071 | 0.088 | 0.057 | 0.079 | **0.478** | **0.439** |
|  | MASE | 0.638 | 1.161 | 0.725 | 1.748 | 1.247 | 0.510 | 0.721 | 0.918 | 0.509 | 0.290 | **0.651** | **0.611** |

## 4.2 MAIN RESULTS

**In-distribution Forecasting.** We begin with an in-distribution evaluation using a total of **29** datasets from the Monash benchmark (Godahewa et al., 2021). Their training set are included in LOTSA (Woo et al., 2024), holding out the test set which we now use for assessments. Figure 3 summarizes the results based on the aggregated mean absolute error (MAE), in comparison with the

baselines presented in the Monash benchmark and the recently released foundation models: TimesFM (200M) (Das et al., 2024), Chronos family (Ansari et al., 2024): Chronos$_S$ (46M), Chronos$_B$ (200M), Chronos$_L$ (710M), and MOIRAI family (Woo et al., 2024): MOIRAI$_S$ (14M), MOIRAI$_B$ (91M), MOIRAI$_L$ (310M). The evaluation results show that MOIRAI-MOE beats all competitors. In particular, MOIRAI-MOE$_S$ drastically surpasses its dense counterpart MOIRAI$_S$ by 17%, and also outperforms the larger models MOIRAI$_B$ and MOIRAI$_L$ by 8% and 7%, respectively. MOIRAI-MOE$_B$ delivers a further 3% improvement over MOIRAI-MOE$_S$. Compared to the foundation model Chronos, which MOIRAI could not surpass, MOIRAI-MOE successfully bridges the gap and delivers superior results with up to 65× fewer activated parameters.

**Zero-shot Forecasting.** Next, we conduct an out-of-distribution evaluation on **10** datasets not included in LOTSA. To establish a comprehensive comparison, we report results for both probabilistic and point forecasting, using continuous ranked probability score (CRPS) and mean absolute scaled error (MASE) as evaluation metrics (see more metrics in Table 8). For baselines, we compare against foundation models TimesFM, TTM (Ekambaram et al., 2024), Timer (Liu et al., 2024c), Moment (Goswami et al., 2024), Time-MoE (Shi et al., 2024), Chronos, and MOIRAI, as well as state-of-the-art full-shot models trained on individual datasets: TiDE (Das et al., 2023), PatchTST (Nie et al., 2023), iTransformer (Liu et al., 2024b), and MoLE-DLinear (Ni et al., 2024). The results are presented in Table 2. **MOIRAI-MOE$_B$ achieves the best overall zero-shot performance, outperforming TimesFM and Chronos that included partial evaluation data in their pretraining corpora**. When compared to all sizes of MOIRAI, MOIRAI-MOE$_S$ delivers a 3%–14% improvement in CRPS and an 8%–16% improvement in MASE. These improvements are remarkable, considering that MOIRAI-MOE$_S$ has only 11M activated parameters – 28× fewer than MOIRAI$_L$.

**Summary.** Our extensive evaluation validates the effectiveness of MOIRAI-MOE's overall model design, demonstrates the strong generalization ability of MOIRAI-MOE, and emphasizes the superiority of token-level specialization over frequency-level approaches (TimesFM, MOIRAI) and models without a specialization module (Chronos). MOIRAI-MOE also performs significantly better than full-shot models trained on each dataset, showing the exceptional capabilities of foundation models.

## 4.3 ABLATION STUDIES

**Model Design.** In the main results, we simultaneously enable the mixture of experts and switch the training objective from a masked encoder approach to a decoder-only approach. To ensure a more rigorous comparison, we conduct further experiments where only the learning objective is changed. Table 3 presents the Monash

Table 3: Model variants performance on Monash.

| Model Variant | Aggregated MAE |
| --- | --- |
| Multi Projection w/ Masked Encoder | 0.78 |
| Multi Projection w/ Decoder-Only | 0.75 |
| Single Projection & MoE w/ Decoder-Only | 0.65 |

evaluation results using the small model, with the first and last rows representing MOIRAI$_S$ and MOIRAI-MOE$_S$, respectively. This outcome suggests that altering the learning objective alone yields modest performance improvements, while the major gains stem from leveraging experts for automatic token-level specialization.

**Training Objective.** We adopt the decoder-only training objective for its superior training efficiency compared to the masked encoder approach. To illustrate this, we conduct experiments with varying

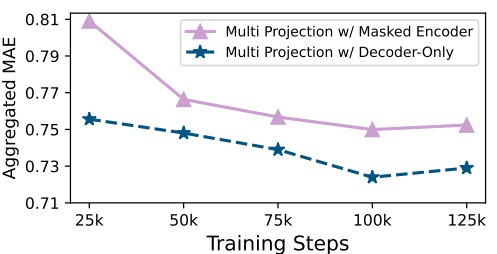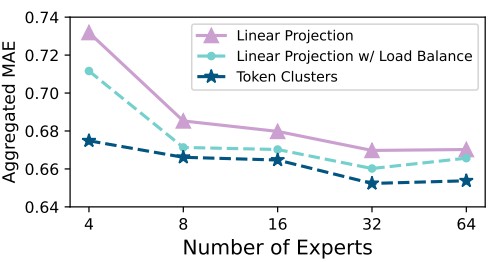

Figure 4: Ablation studies of the training objective and gating function using MOIRAI-MOE$_S$.

training steps, as shown in Figure 4 (left). The results show that the decoder-only approach consistently outperforms the masked encoder at each evaluated step. Moreover, decoder-only training with 50k steps achieves comparable performance to masked encoder training with 100k steps, highlighting the substantial efficiency gains provided by the decoder-only training objective.

**Gating Function.** In Figure 4 (right), we vary the total number of experts and examine the impact of different gating functions on performance. Across all gating functions, performance consistently improves as the number of experts increases. Notably, our proposed token clustering method proves to be consistently superior to the other gating function variants across all expert configurations. This indicates that the clustering approach aligns more closely with the inherent distribution of time series representations that have been optimized in pretraining, leading to more effective expert specialization compared to randomly learned-from-scratch gating. See more results in Appendix B.4.

## 4.4 MODEL ANALYSES

In this section, we delve deeper into the learned token embeddings and expert assignment distribution of MOIRAI-MOE to shed light on the inner workings of the time series MoE foundation model.

**Obs 1: MOIRAI-MOE produces token embeddings in a data-driven way, effectively improving performance.** In Figure 5, we utilize the T-SNE visualization tool (Van der Maaten & Hinton, 2008) to compare the token embeddings generated from the input projection layers of MOIRAI and MOIRAI-MOE. (1) In the first row, we examine the NN5 Daily and Traffic Hourly datasets, which have different frequencies but exhibit similar underlying patterns (visualizations of these patterns can be found in Appendix D). The figure illustrates that MOIRAI produces distinct embeddings due to the use of separate frequency projection layers, while MOIRAI-MOE successfully blends their representations together. Their inherent similarities are further demonstrated by their comparable expert allocation distributions in the last two columns. (2) In the second row, we analyze another daily frequency dataset, Covid Daily Deaths, which shows distinct patterns compared to NN5 Daily. We observe that the embeddings of these two datasets overlap to some extent in the MOIRAI model but are effectively separated in MOIRAI-MOE. Furthermore, the Covid Daily dataset shows different expert selection choices than NN5 Daily due to different token embeddings. **The data-driven modeling paradigm of MOIRAI-MOE ultimately leads to significant performance boosts**, reducing the MAE of NN5 Daily from 5.37 to 4.04 (a 25% improvement), the MAE of Traffic Hourly from 0.02 to 0.013 (a 35% improvement), and the MAE of Covid Daily Deaths from 124.32 to 119 (a 4% improvement).

**Obs 2: Different frequency data exhibit different expert selection distributions at shallow layers but similar distributions at deep layers.** We present the expert allocation distributions on the Monash benchmark grouped by frequency in Figure 6. In the shallow layers, expert selection is

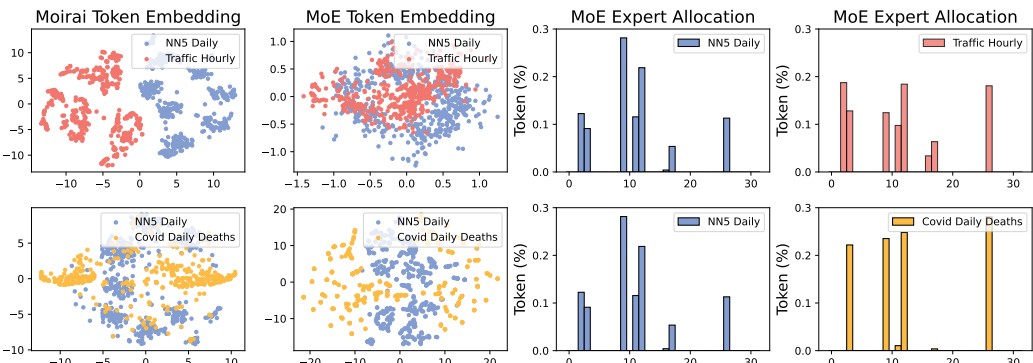

Figure 5: The first two columns are the comparison of embeddings from MOIRAI$_S$ and MOIRAI-MOE$_S$. The last two columns are the expert assignment distributions of MOIRAI-MOE$_S$ in layer 1: the x-axis corresponds to the 32 experts in a layer, and the y-axis is the proportion of tokens that choose experts.

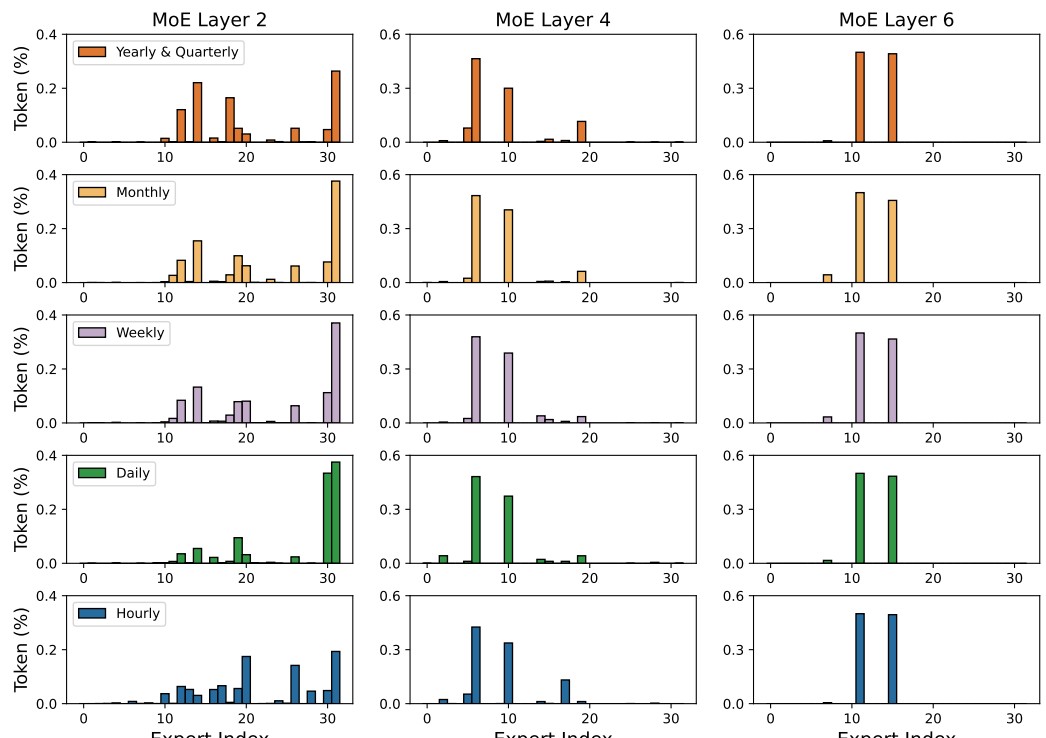

Figure 6: Visualization of the distribution of expert allocation for MOIRAI-MOE$_S$ layers 2, 4, and 6 (the last layer) using the Monash benchmark grouped by time series frequency.

notably diverse, indicating that the model relies on multiple experts to manage the high level of short-term variability, such as cyclical, seasonal, or abrupt changes. As tokens are aggregated in deeper layers, the model shifts its focus to more generalizable temporal dependencies, such as broader trends and long-term patterns, that can be shared across different frequencies and leads to more concentrated experts being selected. By the final layer (layer 6), expert allocation becomes nearly identical across all frequencies, suggesting that the model has abstracted time series into high-level representations largely independent of the frequency. This evidence indicates that **MOIRAI-MOE effectively achieves frequency-invariant hidden representations**, which are crucial for model generalization (Van Ness et al., 2023). The shared parameter space in the last layer also shows that it is sufficient for generating representations needed to make diverse predictions.

**Obs 3: Shallow layers have more routing preferences than deep layers.**    According to Figure 6, as the layer index increases, expert selection gradually converges, with only 3 out of 32 experts being chosen by the final layer. This behavior contrasts with patterns observed in LLMs (Zhu et al., 2024), where earlier layers typically concentrate on a limited number of experts to capture common linguistic features, while deeper layers target more task-specific characteristics. This divergence may stem from the dynamic and noisier nature of time series tokens, which are generated from small time windows, unlike language tokens derived from a fixed vocabulary. **Our findings suggest that denoising processes occur progressively throughout the model**. This observation aligns with conclusions from GPT4TS (Zhou et al., 2023), which found that as the layer depth increases, token vectors are projected into the low-dimensional top eigenvector space of input patterns. Additionally, we recognize that some experts in MOIRAI-MOE are rarely selected. Pruning these underutilized experts for model compression is left for future work.

**Obs 4: Expert allocation reflects time series periodicity patterns.**    To investigate the relationship between the positions of time series tokens and expert allocations, we use hourly data from the Monash repository with a minimum context length of 1,000 (e.g., the Traffic Hourly dataset). Figure 7 visualizes the expert choices at each token position. In the shallow layers, we observe that expert selection follows periodic patterns, consistent with the actual patterns in the raw data, as shown in Figure 13. This suggests that the model dynamically adapts to the cyclical nature of the traffic data,

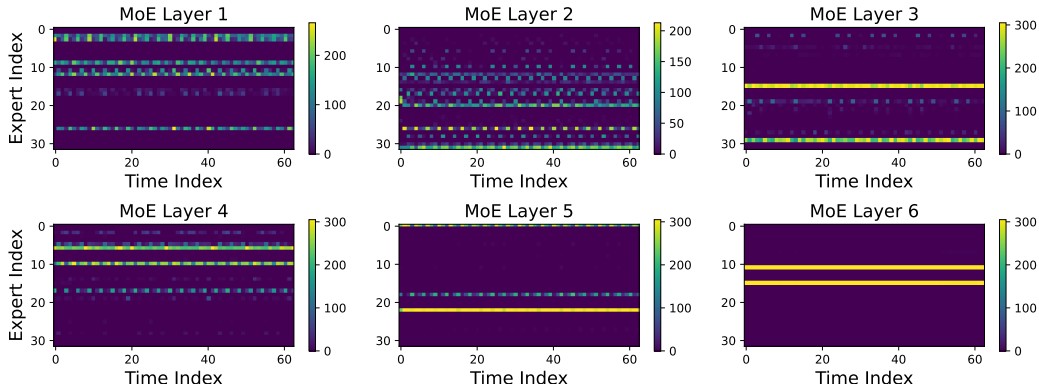

Figure 7: Visualization of expert allocation distributions for MOIRAI-MOE_S. All MoE layers are presented. The x-axis is the time index of the 63 time series tokens, generated from 1,000 context lengths. The y-axis corresponds to the 32 experts in a layer.

assigning specialized experts to manage tokens corresponding to distinct phases of the cycle, such as rising, peaks, and falling. See Appendix B.5 for more details. In short, **MOIRAI-MOE effectively learns to exploit time-based structures and the model specialization operates at the token level**.

## 4.5 EFFICIENCY ANALYSES

In this section, we aim to validate whether the inference speeds of MOIRAI and MOIRAI-MOE are comparable, as we have configured them with similar activated parameters. Additionally, due to the difference in the inference algorithms (the mask encoder in MOIRAI predicts all tokens simultaneously, while the decoder-only approach in MOIRAI-MOE generates predictions autoregressively), we evaluate the inference cost on a subset of the Monash benchmark where the predicted token is one (corresponding to 16 time steps) to eliminate this discrepancy. To also compare to the foundation model Chronos, we set the context length to 512 and the number of sampling samples to 20, aligning with the settings used in Chronos.

We present the summarized results in Table 4 and conclude that MOIRAI-MOE_S and MOIRAI-MOE_B exhibit similar inference times to MOIRAI_S and MOIRAI_B, respectively. These results highlight that MOIRAI-MOE not only maintains the same level of efficiency as MOIRAI but also delivers substantial performance improvements. Additionally, when comparing MOIRAI-MOE to Chronos, which also employs autoregressive inference algorithms, we find that MOIRAI-MOE is significantly faster. This speed advantage stems from the fact that MOIRAI-MOE generates predictions using patches of size 16, while Chronos can be viewed as using a patch size of 1, which greatly affects its inference efficiency.

Table 4: Inference cost evaluation. The values in brackets represent the parameter sizes of the foundation models. For MoE models, the two values indicate the number of activated parameters and the total number of parameters. The spent time is in seconds.

| Model | Chronos_S (46M) | Chronos_B (200M) | Chronos_L (710M) | MOIRAI_S (14M) | MOIRAI_B (91M) | MOIRAI_L (310M) | MOIRAI-MOE_S (11M/117M) | MOIRAI-MOE_B (86M/935M) |
|---|---|---|---|---|---|---|---|---|
| Spent Time (s) | 551 | 1,177 | 2,780 | 264 | 358 | 537 | 273 | 370 |

## 5 CONCLUSION

In this work, we introduce the first time series MoE foundation model MOIRAI-MOE that utilizes sparse experts to model diverse time series patterns in a data-driven manner. Empirical experiments demonstrate that, by enabling automatic token-level specialization, MOIRAI-MOE not only achieves significant performance improvements over all sizes of its predecessor MOIRAI, but also outperforms other competitive foundation models like TimesFM and Chronos with much fewer activated parameters. Moreover, we conduct comprehensive model analyses to gain a deeper understanding of time series MoE foundation models.

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

# A EXPERIMENTAL DETAILS

## A.1 IN-DISTRIBUTION FORECASTING

Following MOIRAI (Woo et al., 2024), we perform evaluations on 29 datasets from the Monash benchmark (Godahewa et al., 2021), including M1 Monthly, M3 Monthly, M3 Other, M4 Monthly, M4 Weekly, M4 Daily, M4 Hourly, Tourism Quarterly, Tourism Monthly, CIF 2016, Australian Electricity Demand, Bitcoin, Pedestrian Counts, Vehicle Trips, KDD Cup 2018, Australia Weather, NN5 Daily, NN5 Weekly, Carparts, FRED-MD, Traffic Hourly, Traffic Weekly, Rideshare, Hospital, COVID Deaths, Temperature Rain, Sunspot, Saugeen River Flow, and US Births. The statistics of data are provided in Table 5, and full results of time series foundation models are shown in Table 6.

Table 5: Summary of datasets used in the in-distribution forecasting evaluations.

| Dataset | Domain | Frequency | Number of Series | Prediction Length |
|---|---|---|---|---|
| M1 Monthly | Econ/Fin | M | 617 | 18 |
| M3 Monthly | Econ/Fin | M | 1,428 | 18 |
| M3 Other | Econ/Fin | M | 174 | 8 |
| M4 Monthly | Econ/Fin | M | 48,000 | 18 |
| M4 Weekly | Econ/Fin | W | 359 | 13 |
| M4 Daily | Econ/Fin | D | 4,227 | 14 |
| M4 Hourly | Econ/Fin | H | 414 | 48 |
| Tourism Quarterly | Econ/Fin | Q | 427 | 8 |
| Tourism Monthly | Econ/Fin | M | 366 | 24 |
| CIF 2016 | Econ/Fin | M | 72 | 12 |
| Aus. Elec. Demand | Energy | 30T | 5 | 336 |
| Bitcoin | Econ/Fin | D | 18 | 30 |
| Pedestrain Counts | Transport | H | 66 | 24 |
| Vehicle Trips | Transport | D | 329 | 30 |
| KDD Cup 2018 | Energy | H | 270 | 168 |
| Australia Weather | Nature | D | 3,010 | 30 |
| NN5 Daily | Econ/Fin | D | 111 | 56 |
| NN5 Weekly | Econ/Fin | W | 111 | 8 |
| Carparts | Sales | M | 2,674 | 12 |
| FRED-MD | Econ/Fin | M | 107 | 12 |
| Traffic Hourly | Transport | H | 862 | 168 |
| Traffic Weekly | Transport | W | 862 | 8 |
| Rideshare | Transport | H | 2,304 | 168 |
| Hospital | Healthcare | M | 767 | 12 |
| COVID Deaths | Healthcare | D | 266 | 30 |
| Temperature Rain | Nature | D | 32,072 | 30 |
| Sunspot | Nature | D | 1 | 30 |
| Saugeen River Flow | Nature | D | 1 | 30 |
| US Births | Healthcare | D | 1 | 30 |

Table 6: Full MAE results of time series foundation models on the Monash Benchmark. The other baseline results can be found in (Woo et al., 2024).

| Dataset | Seasonal Naive | LLMTime | TimesFM | MOIRAI$_{Small}$ | MOIRAI$_{Base}$ | MOIRAI$_{Large}$ | Chronos$_{Small}$ | Chronos$_{Base}$ | Chronos$_{Large}$ | MOIRAI-MOE$_{Small}$ | MOIRAI-MOE$_{Base}$ |
|---|---|---|---|---|---|---|---|---|---|---|---|
| M1 Monthly | 2,011.96 | 2,562.84 | 1,673.60 | 2,082.26 | 2,068.63 | 1,983.18 | 1,797.78 | 1,637.68 | 1,627.11 | 1,992.49 | 1,811.94 |
| M3 Monthly | 788.95 | 877.97 | 653.57 | 713.41 | 658.17 | 664.03 | 644.38 | 622.27 | 619.79 | 646.07 | 617.31 |
| M3 Other | 375.13 | 300.30 | 207.23 | 263.54 | 198.62 | 202.41 | 196.59 | 191.80 | 205.93 | 185.89 | 179.92 |
| M4 Monthly | 700.24 | 728.27 | 580.20 | 597.60 | 592.09 | 584.36 | 592.85 | 598.46 | 584.78 | 569.25 | 544.08 |
| M4 Weekly | 347.99 | 518.44 | 285.89 | 339.76 | 328.08 | 301.52 | 264.56 | 252.26 | 248.89 | 302.65 | 278.37 |
| M4 Daily | 180.83 | 266.52 | 172.98 | 189.10 | 192.66 | 189.78 | 169.91 | 177.49 | 168.41 | 172.45 | 163.40 |
| M4 Hourly | 353.86 | 576.06 | 196.20 | 268.04 | 209.87 | 197.79 | 214.18 | 230.70 | 201.14 | 241.58 | 217.35 |
| Tourism Quarterly | 11,405.45 | 16,918.86 | 10,568.92 | 18,352.44 | 17,196.86 | 15,820.02 | 7,823.27 | 8,835.52 | 8,521.70 | 9,508.07 | 7,374.27 |
| Tourism Monthly | 1,980.21 | 5,608.61 | 2,422.01 | 3,569.85 | 2,862.06 | 2,688.55 | 2,465.10 | 2,358.67 | 2,140.73 | 2,523.66 | 2,268.31 |
| CIF 2016 | 743,512.31 | 599,313.84 | 819,922.44 | 655,888.58 | 539,222.03 | 695,156.92 | 649,110.99 | 604,088.54 | 728,981.15 | 453,631.21 | 568,283.48 |
| Aus. Elec. Demand | 455.96 | 760.81 | 525.73 | 266.57 | 201.39 | 177.68 | 267.18 | 236.27 | 330.04 | 215.28 | 227.92 |
| Bitcoin | 7.78E+17 | 1.74E+18 | 7.78E+17 | 1.76E+18 | 1.62E+18 | 1.87E+18 | 2.34E+18 | 2.27E+18 | 1.88E+18 | 1.55E+18 | 1.90E+18 |
| Pedestrian Counts | 65.60 | 97.77 | 45.03 | 54.88 | 54.08 | 41.66 | 29.77 | 27.34 | 26.95 | 41.35 | 32.37 |
| Vehicle Trips | 32.48 | 31.48 | 21.93 | 24.46 | 23.17 | 21.85 | 19.38 | 19.25 | 19.19 | 21.62 | 21.65 |
| KDD Cup 2018 | 47.09 | 42.72 | 40.86 | 39.81 | 38.66 | 39.09 | 38.60 | 42.36 | 38.83 | 40.21 | 40.86 |
| Australia Weather | 2.36 | 2.17 | 2.07 | 1.96 | 1.80 | 1.75 | 1.96 | 1.84 | 1.85 | 1.76 | 1.75 |
| NN5 Daily | 8.26 | 7.10 | 3.85 | 5.37 | 4.26 | 3.77 | 3.83 | 3.67 | 3.53 | 4.04 | 3.49 |
| NN5 Weekly | 16.71 | 15.76 | 15.09 | 15.07 | 16.42 | 15.30 | 15.03 | 15.12 | 15.09 | 15.74 | 15.29 |
| Carparts | 0.67 | 0.44 | 0.50 | 0.53 | 0.47 | 0.49 | 0.52 | 0.54 | 0.53 | 0.45 | 0.44 |
| FRED-MD | 5,385.53 | 2,804.64 | 2,237.63 | 2,568.48 | 2,679.29 | 2,792.53 | 938.46 | 1,036.67 | 863.99 | 1,651.76 | 2,273.61 |
| Traffic Hourly | 0.013 | 0.030 | 0.009 | 0.020 | 0.020 | 0.010 | 0.013 | 0.012 | 0.010 | 0.013 | 0.014 |
| Traffic Weekly | 1.19 | 1.15 | 1.06 | 1.17 | 1.14 | 1.13 | 1.14 | 1.12 | 1.12 | 1.13 | 1.14 |
| Rideshare | 1.60 | 6.28 | 1.36 | 1.35 | 1.39 | 1.29 | 1.27 | 1.33 | 1.30 | 1.26 | 1.26 |
| Hospital | 20.01 | 25.68 | 18.54 | 23.00 | 19.40 | 19.44 | 19.74 | 19.75 | 19.88 | 20.17 | 19.60 |
| COVID Deaths | 353.71 | 653.31 | 623.47 | 124.32 | 126.11 | 117.11 | 207.47 | 118.26 | 190.01 | 119.00 | 102.92 |
| Temperature Rain | 9.39 | 6.37 | 5.27 | 5.30 | 5.08 | 5.27 | 5.35 | 5.17 | 5.19 | 5.33 | 5.36 |
| Sunspot | 3.93 | 5.07 | 1.07 | 0.11 | 0.08 | 0.13 | 0.20 | 2.45 | 3.45 | 0.10 | 0.08 |
| Saugeen River Flow | 21.50 | 34.84 | 25.16 | 24.07 | 24.40 | 24.76 | 23.57 | 25.54 | 26.25 | 23.05 | 24.40 |
| US Births | 1,152.67 | 1,374.99 | 461.58 | 872.51 | 624.30 | 476.50 | 432.14 | 420.08 | 432.14 | 411.61 | 385.24 |

## A.2 ZERO-SHOT FORECASTING

We conduct zero-shot evaluations on the datasets listed in Table 7, which cover five domains and span frequencies ranging from minute-level to weekly. We use a non-overlapping rolling window approach, where the stride equals the prediction length. The test set consists of the last $h * r$ time steps, where $h$ is the forecast horizon and $r$ is the number of rolling evaluation windows. The validation set is defined as the last forecast horizon before the test set, while the training set includes all preceding data. The zero-shot performance measured by MSE and MAE is provided in Table 8.

Table 7: Summary of datasets used in the zero-shot forecasting evaluations.

| Dataset | Domain | Frequency | Prediction Length | Rolling Evaluations |
|---|---|---|---|---|
| Electricity (Trindade, 2015) | Energy | H | 24 | 7 |
| Solar (Lai et al., 2018) | Energy | H | 24 | 7 |
| Turkey Power [1] | Energy | H | 24 | 7 |
| ETT1 (Zhou et al., 2021) | Energy | D | 30 | 3 |
| ETT2 (Zhou et al., 2021) | Energy | D | 30 | 3 |
| Istanbul Traffic [2] | Transport | H | 24 | 7 |
| M-DENSE (Jiang et al., 2023) | Transport | D | 30 | 3 |
| Walmart (Walmart Competition Admin, 2014) | Sales | W | 8 | 4 |
| Jena Weather (Wu et al., 2021) | Nature | 10T | 144 | 7 |
| BizITObs-L2C (Palaskar et al., 2024) | Web/CloudOps | 5T | 48 | 20 |

Table 8: Zero-shot forecasting performance measured by MSE and MAE. Asterisks (*) indicate the non-zero-shot datasets. The Avg column is normalized by seasonal naive, followed by geometric mean. Two Avg values are shown: one that averages all data, and another (non-leak) excludes Electricity and Solar. Best average results are highlighted in **red**, and second best results are in **blue**. Power: Turkey Power. Traffic: Istanbul Traffic. Weather: Jena Weather. BizITObs: BizITObs-L2C.

| Method | Metric | Electricity | Solar | Power | ETT1 | ETT2 | Traffic | MDENSE | Walmart | Weather | BizITObs | Avg (all) | Avg (non-leak) |
|---|---|---|---|---|---|---|---|---|---|---|---|---|---|
| Seasonal Naive | MSE | 1299429.16 | 1293.24 | 1798196.83 | 57976.63 | 122878.95 | 203.32 | 39929.67 | 32876026.66 | 2197.23 | 174.31 | 1.000 | 1.000 |
| | MAE | 166.20 | 15.77 | 492.60 | 154.98 | 211.56 | 8.72 | 118.38 | 2637.43 | 10.96 | 9.69 | 1.000 | 1.000 |
| iTransformer | MSE | 1264494.38 | 1183.57 | 968959.56 | 55320.57 | 178757.02 | 41.77 | 9905.39 | 10922819.00 | 1885.01 | 20.55 | 0.508 | 0.435 |
| | MAE | 165.89 | 17.61 | 399.09 | 170.83 | 279.21 | 4.85 | 51.06 | 1560.68 | 10.65 | 2.66 | 0.741 | 0.678 |
| MoLE-DLinear | MSE | 1901617.97 | 1098.56 | 1071490.46 | 39026.37 | 195287.19 | 153.71 | 13016.78 | 26832049.08 | 1649.90 | 21.57 | 0.656 | 0.575 |
| | MAE | 197.06 | 16.47 | 420.67 | 130.79 | 328.28 | 8.48 | 62.43 | 2395.50 | 12.81 | 2.75 | 0.857 | 0.803 |
| TimesFM | MSE | 1378828.95* | 1061.70 | 384815.80 | 42789.02 | 169714.41 | 106.01 | 10194.73 | 9494507.86 | 1317.09 | 23.23 | 0.475 | 0.401 |
| | MAE | 137.57* | 18.07 | 277.94 | 138.42 | 245.61 | 5.75 | 49.78 | 1484.68 | 7.94 | 2.89 | 0.672 | **0.612** |
| TTM | MSE | 2432897.66 | 884.33* | 647289.67 | 56256.46 | 116203.30 | 114.79 | 18425.62 | 39297380.00 | 1122.55 | 23.41 | 0.625 | 0.538 |
| | MAE | 179.56 | 16.46* | 341.96 | 158.85 | 213.61 | 7.53 | 86.44 | 3360.79 | 8.88 | 2.97 | 0.833 | 0.784 |
| Timer | MSE | 2205084.30 | 962.26 | 687600.25 | 39235.36 | 129063.67 | 75.23 | 19875.60 | 29410540.00 | 1873.68 | 27.21 | 0.613 | 0.527 |
| | MAE | 200.62 | 17.57 | 370.53 | 131.31 | 235.27 | 6.42 | 87.72 | 2646.92 | 13.65 | 3.50 | 0.865 | 0.804 |
| Moment | MSE | 44303358.90 | 2876.47 | 3272382.39 | 46075.47 | 411967.28 | 601.62 | 19506.54 | 29046437.85 | 1804.48 | 129.26 | 1.760 | 1.180 |
| | MAE | 843.50 | 41.02 | 873.48 | 152.56 | 484.86 | 21.87 | 90.51 | 2690.84 | 16.89 | 9.11 | 1.650 | 1.355 |
| Chronos$_S$ | MSE | 1251170.49* | 1405.10* | 418195.72 | 60157.02 | 112472.02 | 100.62 | 15377.29 | 14697271.28 | 3945.04 | 23.89 | 0.587 | 0.511 |
| | MAE | 126.25* | 15.79* | 275.11 | 161.23 | 207.11 | 5.28 | 59.26 | 1693.33 | 16.90 | 2.94 | 0.724 | 0.691 |
| Chronos$_B$ | MSE | 1147348.35* | 1062.73* | 400709.37 | 66320.26 | 107178.21 | 80.48 | 12770.66 | 15813384.14 | 1720.53 | 22.78 | 0.501 | 0.439 |
| | MAE | 121.69* | 13.18* | 285.79 | 169.60 | 194.70 | 4.69 | 51.58 | 1706.11 | 10.28 | 2.82 | 0.656 | 0.628 |
| Chronos$_L$ | MSE | 1073679.39* | 1017.98* | 362386.33 | 73974.48 | 106362.90 | 98.20 | 13625.07 | 12339319.84 | 1874.83 | 23.61 | 0.503 | 0.447 |
| | MAE | 121.06* | 12.86* | 277.64 | 177.68 | 191.97 | 5.07 | 53.61 | 1560.11 | 11.30 | 2.89 | 0.664 | 0.639 |
| MOIRAI$_S$ | MSE | 4015423.50 | 1429.82 | 757613.06 | 39481.46 | 118636.33 | 146.24 | 11041.41 | 19886286.00 | 1932.16 | 22.48 | 0.647 | 0.498 |
| | MAE | 219.02 | 19.19 | 358.01 | 133.82 | 209.68 | 8.71 | 58.25 | 2112.07 | 10.23 | 2.90 | 0.802 | 0.715 |
| MOIRAI$_B$ | MSE | 1734656.25 | 1105.95 | 477193.47 | 51793.64 | 113074.23 | 44.60 | 17724.71 | 18981036.00 | 1196.21 | 22.44 | 0.500 | 0.414 |
| | MAE | 164.94 | 16.97 | 293.74 | 149.15 | 202.89 | 4.72 | 79.41 | 2046.22 | 7.73 | 2.81 | 0.713 | 0.650 |
| MOIRAI$_L$ | MSE | 1229872.00 | 997.13 | 340307.44 | 44752.48 | 106513.38 | 101.17 | 14874.89 | 21274000.00 | 1914.39 | 21.79 | 0.511 | 0.449 |
| | MAE | 150.66 | 16.25 | 262.70 | 142.21 | 204.72 | 5.93 | 69.73 | 2110.73 | 10.10 | 2.77 | 0.720 | 0.669 |
| Time-MoE$_B$ | MSE | 1158323.38* | 176.27* | 315704.91 | 50267.22 | 114374.42 | 89.87 | 11303.31 | 13934856.92 | 1371.87 | 28.51 | **0.395** | 0.408 |
| | MAE | 120.52* | 7.07* | 254.28 | 149.21 | 218.55 | 5.70 | 57.43 | 1742.96 | 11.35 | 3.26 | **0.644** | 0.663 |
| Time-MoE$_L$ | MSE | 1203643.75* | 194.84* | 350989.67 | 47389.70 | 121112.59 | 99.13 | 9585.73 | 12876789.32 | 1264.26 | 27.34 | **0.394** | 0.400 |
| | MAE | 120.53* | 9.06* | 262.48 | 147.11 | 229.67 | 6.45 | 52.10 | 1687.08 | 9.32 | 3.24 | 0.650 | 0.652 |
| MOIRAI-MoE$_S$ | MSE | 930140.63 | 1113.50 | 360995.59 | 45412.81 | 114609.09 | 53.05 | 9426.45 | 18025986.00 | 1944.27 | 23.45 | 0.453 | **0.395** |
| | MAE | 138.03 | 16.05 | 260.82 | 141.08 | 194.63 | 4.78 | 50.09 | 1955.77 | 10.08 | 2.89 | 0.668 | 0.617 |
| MOIRAI-MoE$_B$ | MSE | 907276.31 | 1047.63 | 311227.06 | 48487.21 | 107284.42 | 45.83 | 9740.51 | 17094764.00 | 1954.24 | 22.54 | 0.434 | **0.378** |
| | MAE | 122.27 | 15.24 | 251.10 | 145.50 | 191.47 | 4.33 | 49.73 | 1919.31 | 10.31 | 2.80 | **0.646** | **0.605** |

## A.3 METHODS

The following is a brief introduction to the models used in the evaluation process.

---

[1]https://www.kaggle.com/datasets/dharanikra/electrical-power-demand-in-turkey
[2]https://www.kaggle.com/datasets/leonardo00/istanbul-traffic-index

- TiDE (Das et al., 2023) encodes the historical data of a time series along with covariates using dense multi-layer perceptrons (MLPs). It then decodes the time series while incorporating future covariates, also utilizing dense MLPs for this process.
- PatchTST (Nie et al., 2023) employs Transformer encoders combined with patching and channel independence techniques to enhance the performance of time series forecasting.
- iTransformer (Liu et al., 2024b) treats independent time series as tokens to effectively capture multivariate correlations through self-attention.
- MoLE-DLinear (Ni et al., 2024) trains multiple linear-centric models (i.e., experts) and a router model that weighs and mixes their outputs. In this study, we use the DLinear model as the experts.
- LLMTime (Gruver et al., 2023) is a method for time series forecasting that leverages Large Language Models by encoding numerical data as text and generating possible future values through text completions.
- TimesFM (Das et al., 2024) is a decoder-only time series foundation model that pretrained on a large corpus of time series data, including both real-world and synthetic datasets.
- TTM (Ekambaram et al., 2024) is a foundation model based on the light-weight TSMixer architecture, incorporating innovations like adaptive patching, diverse resolution sampling, and resolution prefix tuning.
- Timer (Liu et al., 2024c) is a decoder-only foundation model, presenting notable few-shot generalization, scalability, and task generality.
- Moment (Goswami et al., 2024) refers to a family of open time series foundation models that canhandle different time series analysis tasks.
- Chronos (Ansari et al., 2024) is an encoder-decoder time series foundation model that uses quantization to convert real numbers into discrete tokens.
- MOIRAI (Woo et al., 2024) is a time series foundation model trained on the LOTSA dataset, which contains over 27 billion observations across nine diverse domains.
- Time-MoE (Shi et al., 2024) is a concurrent work that applies mixture of experts techniques to time series foundation models.
- MOIRAI-MOE is proposed in this study, which is capable of achieving automatic token-level specialization.

**Context Length Setting for All Methods.** In Table 9, we detail the context lengths used for each method in this study, and in their original paper. For full-shot deep learning models, we believe our searching range generally covers the lengths set in their original paper. For foundation models, the choice of input lengths depends on their pretraining strategies. For instance, in the case of TimesFM and Chronos, the input lengths are consistently set to 512 during pretraining. In contrast, for MOIRAI and MOIRAI-MOE, the pretraining algorithm involves randomly sampling a context length in the range [2, 8192]. Thus, searching for the input length on validation set during inference is needed.

Table 9: Comparisons of methods' context lengths: this study versus original papers.

| Model | In-Dist. Evaluation (29 datasets) | Zero-Shot Evaluation (10 datasets) | Original Paper |
|---|---|---|---|
| TiDE | – | Searching within prediction lengths * [2,20] | 720 |
| PatchTST | – | Searching within prediction lengths * [2,20] | 336 |
| iTransformer | – | Searching within prediction lengths * [2,20] | 96 |
| TTM | – | 512 | 512 |
| Timer | – | 672 | 672 |
| Moment | – | 512 | 512 |
| Time-MoE | – | 4,096 | {512, 1024, 2048, 3072} |
| TimesFM | 512 | 512 | 512 |
| Chronos | 512 | 512 | 512 |
| MOIRAI | 1000 | Searching within range {1000, 2000, 3000, 4000, 5000} | Searching within range {1000, 2000, 3000, 4000, 5000} |
| MOIRAI-MOE | 1000 | Searching within range {1000, 2000, 3000, 4000, 5000} | Searching within range {1000, 2000, 3000, 4000, 5000} |

**Hyperparameter Search for Full-Shot Methods.** For the three full-shot models used in zero-shot forecasting part, i.e., TiDE (Das et al., 2023), PatchTST (Nie et al., 2023), and iTransformer (Liu et al., 2024b), we conduct hyperparameter search based on the values specified in Table 10. In addition, we explore the learning rate in the range [1e-6, 1e-3] on a log scale, and set the context

length as $l = m * h$, where $m$ is tuned in the range [2, 20], and $h$ is the prediction length. We implement a random search across these parameters over 15 training runs and report results based on the best validation CRPS.

Table 10: Hyperparameter search values for TiDE, PatchTST, and iTransformer.

|  | **Hyperparameter** | **Values** |
|---|---|---|
| TiDE | hidden_dim | {64, 128, 256} |
|  | num_encoder_layers | [2, 6] |
|  | num_decoder_layers | [2, 6] |
| PatchTST | d_model | {64, 128, 256} |
|  | num_encoder_layers | [2,6] |
| iTransformer | d_model | {128, 256, 512} |
|  | num_encoder_layers | [2, 4] |

**MOIRAI-MOE Training Details.** All MOIRAI-MOE models are trained on 16 A100 (40G) GPUs using a batch size of 1,024 and bfloat16 precision. The small and base model are trained for 50,000 and 250,000 steps on LOTSA (Woo et al., 2024), respectively. The patch size $P$ is set to 16 and the masking ratio $r$ for decoder-only training is 0.3 (the corresponding experiments are provided in Appendix B). For optimization, we utilize the AdamW optimizer with lr = 1e-3, weight decay = 1e-1, $\beta_1 = 0.9$, $\beta_2 = 0.98$. We also apply a learning rate scheduler with linear warmup for the first 10,000 steps, followed by cosine annealing.

## B  ADDITIONAL RESULTS

### B.1  EFFECTS OF TRAINING STEPS

In Figure 8, we present a comparison between MOIRAI$_S$ and MOIRAI-MOE$_S$ in terms of training steps. The results demonstrate that MOIRAI-MOE outperforms MOIRAI from the very first evaluation point – 25k steps. Furthermore, MOIRAI-MOE at 25k steps achieves better performance than MOIRAI at 125k steps. This figure highlights the clear advantages of MOIRAI-MOE in terms of both model performance and reduced training steps.

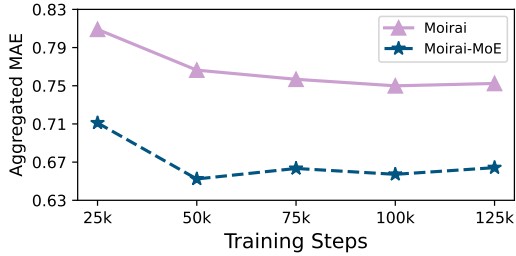

Figure 8: Performance comparison between MOIRAI and MOIRAI-MOE across training steps.

### B.2  EFFECTS OF PATCH SIZE

In contrast to MOIRAI, which designs multiple input/output projection layers, each associated with a specific patch size, MOIRAI-MOE utilizes a single projection layer with a single patch size. In this part, we conduct experiments to examine the impact of different patch size choices. The evaluation results on the Monash benchmark are presented in Figure 9 (left), where the patch size of 16 yields the best performance. Increasing or decreasing this size results in performance degradation. Additionally, patch size affects inference speed; with a fixed context window, smaller patch sizes generate more time series tokens, increasing GPU memory usage and ultimately slowing down inference. For instance, using a patch size of 4 can take over a day to complete all evaluations. Our choice of a patch size of 16 not only delivers strong performance but also maintains a reasonable inference speed.

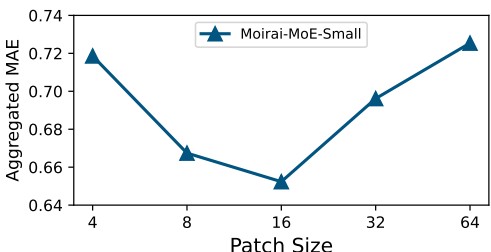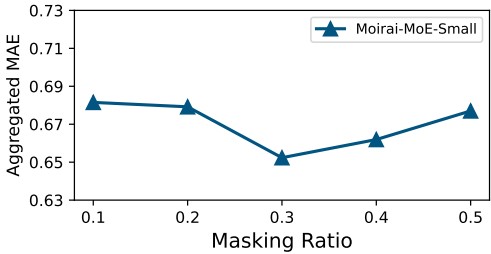

Figure 9: Effects of patch size and masking ratio using MOIRAI-MOE$_S$.

### B.3  EFFECTS OF MASKING RATIO

In this study, we introduce the masking ratio $r$ as a hyperparameter that determines the portion of the entire sequence used solely for robust normalizer calculation, helping to mitigate distribution shift issues. We conduct experiments to assess the effects of different masking ratios, with the evaluation results on the Monash benchmark shown in Figure 9 (right). A masking ratio of 0.3 delivers the best performance. A ratio of 0.1 uses too little data to compute a robust normalizer, potentially failing to accurately represent the overall sequence statistics. Conversely, a ratio of 0.5 masks half of the data, which may hinder the parallel learning efficiency in decoder-only training. Therefore, it is crucial to select an appropriate data range that is small enough to avoid excessive masking, yet sufficiently representative for robust normalizer computation.

### B.4  EXPERT DISTRIBUTIONS OF DIFFERENT GATING FUNCTION

In this part, we present an in-depth comparison of the different gating functions explored in this study.

First, we provide additional details on the implementation of the proposed token clustering method. The core idea of this approach is to leverage cluster centroids derived from the token representations of a pretrained model to guide expert allocations. Specifically, we perform inference on our training corpus, LOTSA, using data amount corresponding to 100 epochs. During this process, we extract the self-attention output representations from a pretrained MOIRAI model and apply mini-batch k-means clustering to continuously update the clusters. The number of clusters is set to match the total number of experts. During the training of the MoE model, each token computes the Euclidean distance to each cluster centroid, and these distances are used as token-to-expert affinity scores for expert assignments. Empirical evaluations have demonstrated the effectiveness of this approach compared to randomly learned gating from scratch, indicating that the clustering method better aligns with the inherent distribution of time series representations.

Using the three gating functions explored in this study, i.e., linear projection, linear projection with load balancing, and token clustering, we present their expert allocation distributions aggregated across all datasets in the Monash benchmark, as illustrated in Figure 10. In terms of selection diversity, we observe the following relationships: Token Clusters (least diverse) < Pure Linear Projection (neutral) < Linear Projection with Load Balancing (most diverse). According to their performance results shown in Figure 4, we can establish the following ranking: Token Clusters > Linear Projection with Load Balancing > Pure Linear Projection. Based on all these observations, we offer the following explanation:

- In the token clusters approach, the expert selections are less diverse because the routing is grounded in pretrained knowledge. The clustering step creates centroids that represent well-structured patterns in the data, and then tokens are routed to specific experts that are particularly suited to handle the type of data represented by their corresponding cluster. While this targeted routing reduces diversity, it enhances performance due to the selection of experts based on more meaningful criteria.

- The addition of load balancing loss increases the diversity of expert selection by spreading the workload and encouraging the use of all experts more evenly. This diversity prevents over-reliance on specific experts, potentially improving generalization and performance compared to pure linear

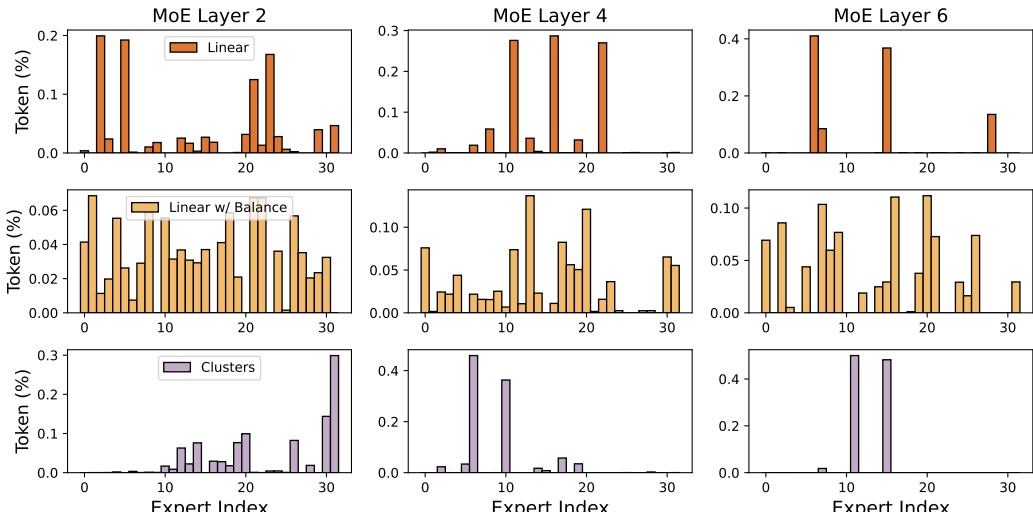

Figure 10: Visualization of the distribution of expert allocation for MOIRAI-MOE$_S$ layers 2, 4, and 6 (the last layer) using all data from the Monash benchmark.

projection. However, this approach might be less targeted than clustering, since it still depends on a learned gating function rather than pretrained centroids.

- In the pure linear projection method, the gating function is entirely learned from scratch. Without any additional constraints (like load balancing), certain experts might get selected more often than others, leading to a neutral level of diversity. Since there is no mechanism to encourage exploration (like load balancing) or specialized routing (like clustering), performance remains lower than the other methods.

## B.5 VISUALIZATION OF TIME SERIES OBSERVATIONS AND EXPERT ALLOCATIONS

Following the discussion in the main paper, this section investigates the relationship between raw time series observations and their corresponding expert allocations. In Figure 11, the upper subfigure presents a Traffic Hourly time series sequence with a length of 512. For enhanced visualization, the sequence is segmented using vertical dashed lines, each spanning 16 steps, which is equal to the length of a single time series token. The lower subfigure illustrates the expert allocations at shallow layers for 32 tokens derived from the 512 observations. The yellow straight line represents the specific experts selected by the token at each position. The alignment of subfigures facilitates an intuitive comparison between the time series trends and the associated expert selections.

The figure includes red square boxes to highlight time series segments exhibiting a downward trend followed by a slight upward pattern. These segments consistently correspond to the activation of two specific experts, as shown in the lower subfigure. This observation suggests that Moirai-MoE effectively captures time-based structures and demonstrates model specialization at the token level.

## C LIMITATION

The limitation of this study lies in the efficiency of autoregressive predictions during inference, a well-documented challenge for decoder-only architectures. However, inference solutions developed for large language models (LLMs) could help address this issue. For instance, many LLMs leverage quantization techniques (e.g., 8-bit or 4-bit weights) to significantly reduce computational costs while maintaining performance. In future work, we plan to explore model quantization and pruning methods to optimize efficiency by removing less critical parameters, such as underutilized experts in deeper layers. Additionally, we aim to implement key-value (KV) caching techniques to accelerate inference. However, a key challenge lies in our use of instance normalization, which requires recalculating

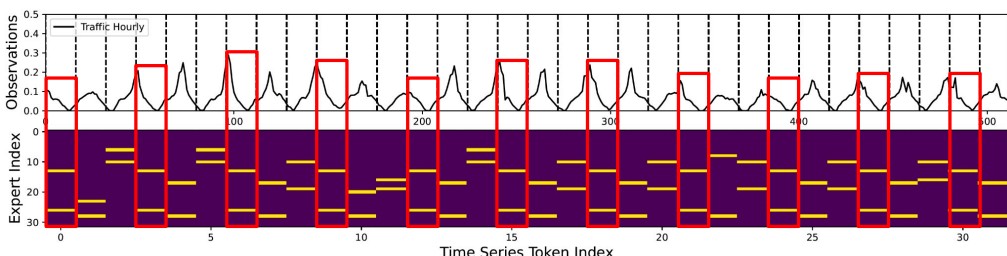

Figure 11: Joint visualization of raw time series observations and their corresponding expert allocation distributions at shallow layers of MOIRAI-MOE$_S$. The upper subfigure depicts the raw time series observations with the x-axis representing time step indices (0 to 511). The lower subfigure shows the expert allocation distributions, where the x-axis corresponds to the time series token indices (0 to 31), and the y-axis represents the indices of the 32 experts in the layer.

normalization statistics whenever a new token is generated. This necessity could render the cached hidden states invalid, presenting an obstacle to efficient caching.

## D  VISUALIZATION

In this section, we visualize the datasets used in the model analyses (NN5 Daily (Figure 12), Traffic Hourly (Figure 13), and Covid Daily Deaths (Figure 14)) to facilitate understanding of the patterns within the time series data.

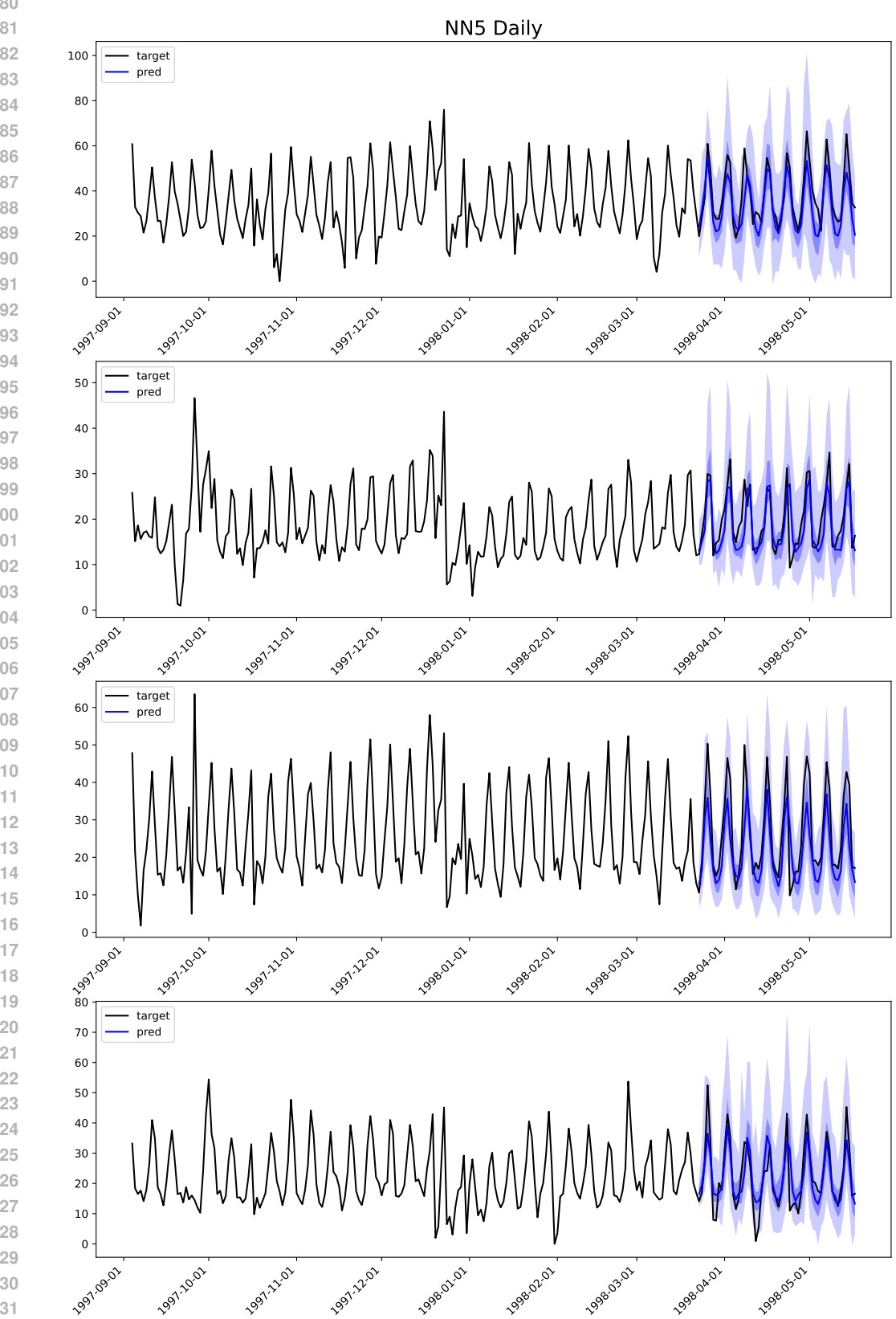

Figure 12: Visualization of NN5 Daily data, including both context length and forecast results.

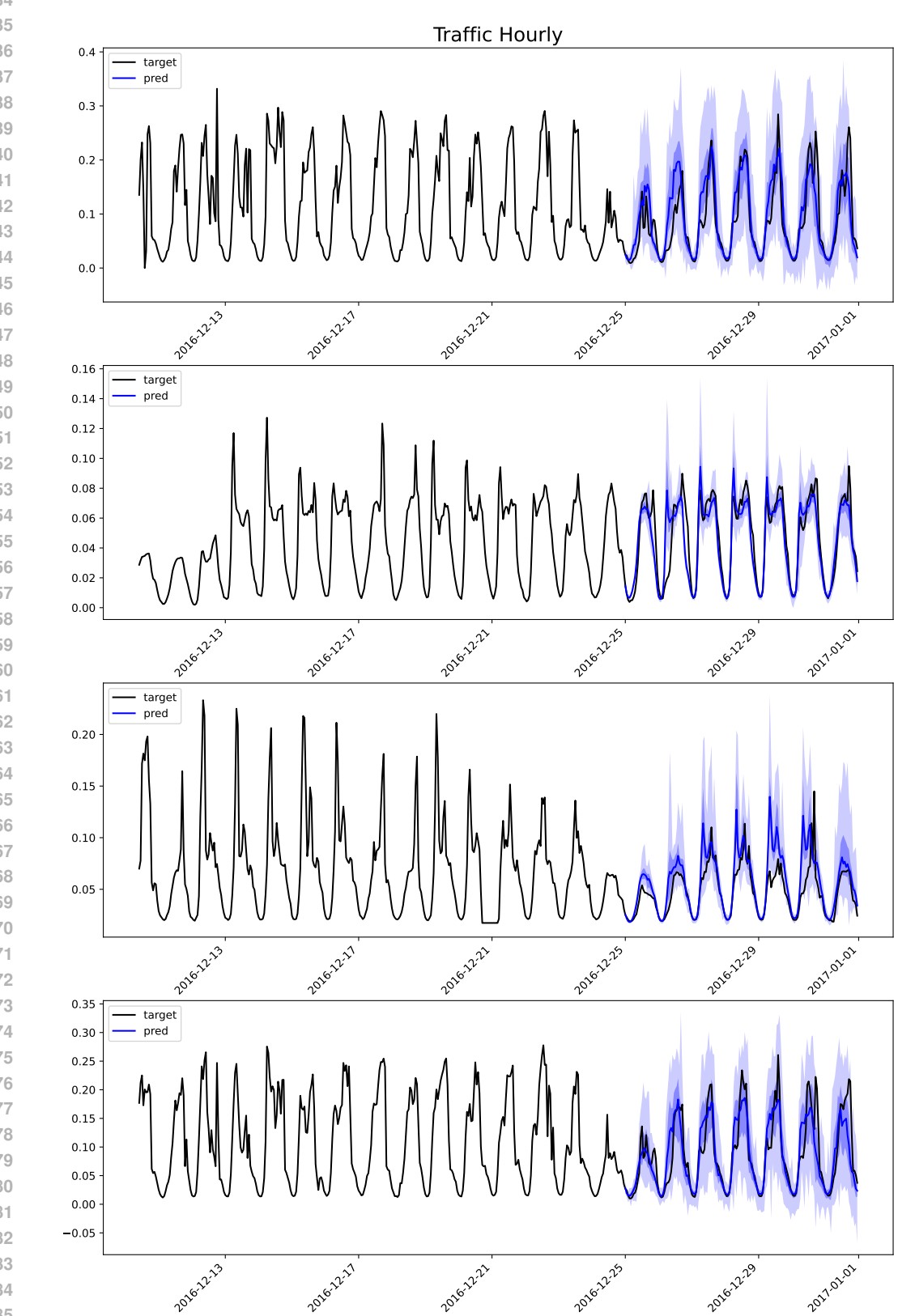

Figure 13: Visualization of Traffic Hourly data, including both context length and forecast results.

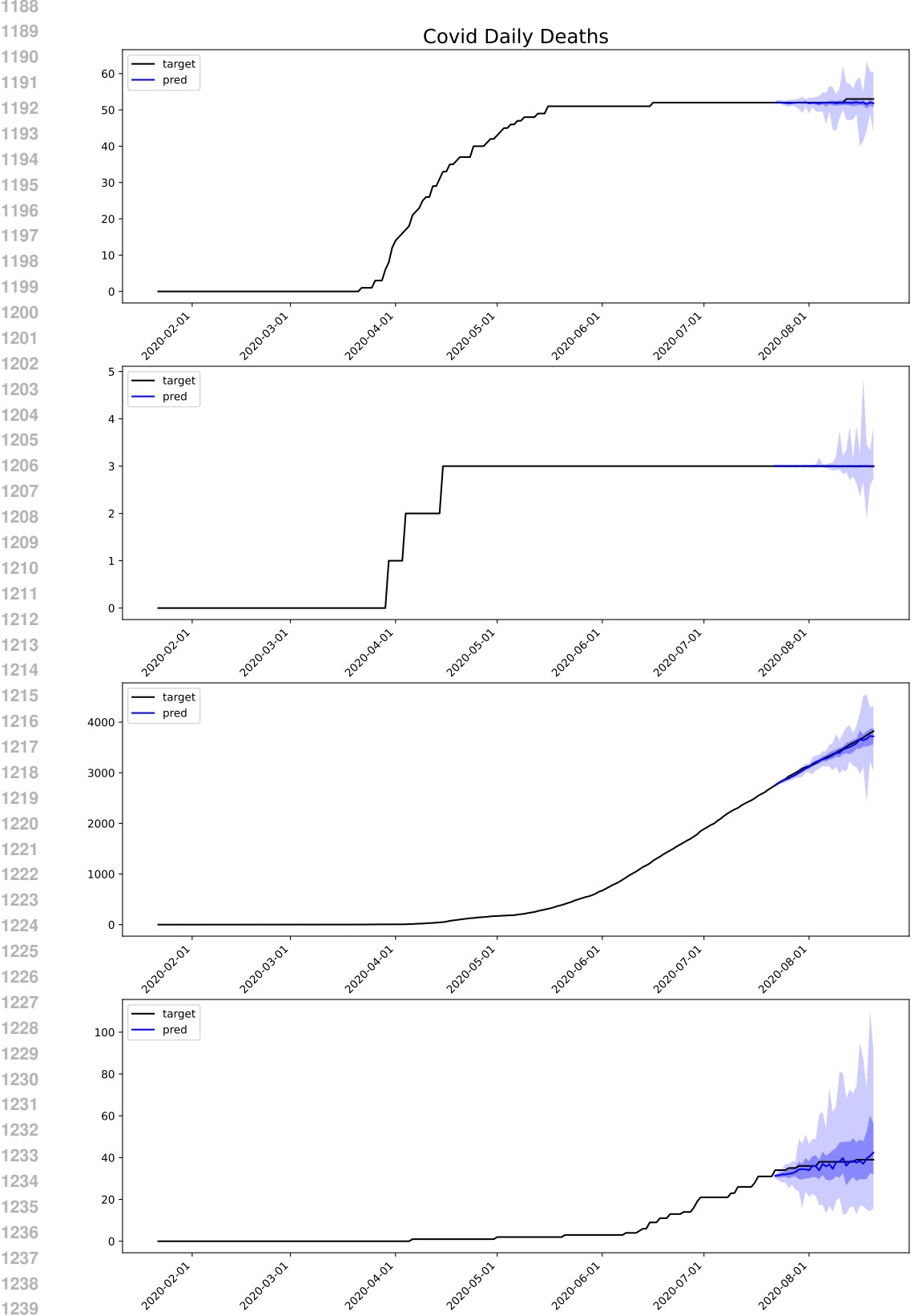

Figure 14: Visualization of Covid Daily Deaths, including both context length and forecast results.

