# OpenReview forum: "Moirai-MoE: Empowering Time Series Foundation Models with Sparse Mixture of Experts"
_ICLR.cc/2025/Conference — Submitted to ICLR 2025_

### Official Review · Reviewer_JYvV · 2024-10-25

**Soundness:** 2
**Presentation:** 2
**Contribution:** 2
**Rating:** 3
**Confidence:** 5

**Summary:**

The paper introduces Moirai-MoE, a time series model that incorporates sparse Mixture of Experts (MoE) for forecasting applications. The authors adapt an existing Transformer-based forecasting model, namely Moirai, by replacing its feed-forward networks (FFNs) with well established MoE layers. Evaluations in both in-distribution and zero-shot settings indicate that the use of MoE layers is beneficial across forecasting applications.

**Strengths:**

1. The paper is well structured.
2. The authors conduct extensive forecasting experiments, including 29 in-distribution and 10 zero-shot applications, to evaluate their method.

**Weaknesses:**

1. Time series models utilising Mixture of Experts (MoE) have been well studied since the 1990s [1] up until now [2,3,4]. As the work under review combines well-established MoE approaches with existing time series models for a specific task, i.e. forecasting, it offers a limited technical contribution to the field of time series analysis.

2. The related work section is incomplete, as the authors fail to discuss existing time series model that utilise MoE. Previous works stated above [1,2,3,4] and further works should be included by the authors to provide a representative overview of the current literature.

3. The methodology section of the paper is poorly elaborated, e.g.:
   - None of the equations (1) to (7) state the dimension of the input and the output variables.
   - While it may refer to a single Transformer layer, the variable $l$ is introduced in equation (1) without further explanation.
   - In equation (5), the load balancing loss is multiplied by $M$ without further explanation, the indicator function $\mathbb{1}$ is introduced without further explanation, and the argmax operation is performed without defining over which set.
   - It is unclear whether the load balancing loss $L_{load}$ is defined for each layer $l \in L$.
   - It is unclear how the load balancing loss $L_{load}$ is combined with the training objective $L_{pred}$ provided in equation (7).
   - It is unclear how the cluster centroids $C \in \mathbb{R}^{M \times D}$ are derived from the self-attention output representations $\tilde{x}^{l-1}$.

4. The experiments should include existing time series models that use MoE, e.g. [3,4], to enable a fair comparison of the proposed MoE approach. Furthermore, the results should be reported over multiple seeds to ensure robustness. Well established metrics such as the mean squared error (MSE) or mean absolute error (MAE) should be reported to enable fair comparison with methods that are not included as baselines.

5. The authors do not elaborate on the limitations of their work.

6. The authors do not provide code to support reproducibility.

[1] Zeevi et al. "Time series prediction using mixtures of experts." NeurIPS (1996).

[2] Yuksel et al. "Twenty years of mixture of experts." IEEE transactions on neural networks and learning systems (2012).

[3] Ni et al. "Mixture-of-Linear-Experts for Long-term Time Series Forecasting." AISTATS (2024).

[4] Shi et al. "Time-MoE: Billion-Scale Time Series Foundation Models with Mixture of Experts." arXiv (2024).

**Questions:**

1. How does the choice of pre-trained model influence the resulting cluster centroids $C \in \mathbb{R}^{M \times D}$ and, consequently, the downstream performance? To this end, have the authors investigated any other pre-trained model besides Moirai?

2. In Table 1, rather than presenting the number of activated paramters, it would be more insightful to show how the downstream performance of Moirai-MoE evolves during training, e.g. with increasing FLOPs and GPU hours, compared to Moirai. Does Moirai-MoE match or even outperform Moirai with less compute? If so, at what point during training do the advantages of the MoE layers become evident?

3. The trends depicted in Figure 4 indicate that it might be worth increasing the training steps ($>$ 100k) and the number of experts ($>$ 32). To this end, could the authors extend their analysis to see whether downstream performance can be further improved?

4. In Figure 5, it seems that experts of the Covid Daily Deaths are a subset of the experts allocated for the Traffic Hourly data. Could the authors explain whether this is based on data similarity or any other phenomenon?

5. In Figure 6, the expert allocation appears very similar across frequencies and is notably sparse, suggesting that some experts might be redundant. How does this align with Figure 4, which indicates that downstream performance improves as the number of experts increases?

6. Furthermore, is it necessary to replace all FFNs with MoE layers? Existing work on MoE in natural language processing [5] and computer vision [6] show that replacing only certain layers has little impact on downstream performance, while saving computation time due to less communication overhead.

7. Have the authors analysed what the experts learn? Do some experts focus on local patterns while others focus on global patterns? Do some experts analyse low-frequency components while others analyse high-frequency components? The authors attempt to explain expert allocation with reference to Figure 7, however, the figure is poorly structured and does not support clear conclusions.

8. The authors provide an efficiency analysis focused on the inference cost of predicting a single token, i.e. a forecasting horizon of 1 token. While this does not allow for a fair evaluation of real-world applications, the authors might rather investigate how inference cost scales with an increasing forecasting horizon.

9. Regarding Figure 2 and Table 3, do the authors plan to provide full results for a fair comparison?

[5] Lepikhin et al. "GShard: Scaling giant models with conditional computation and automatic sharding." ICLR (2021).

[6] Riquelme et al. "Scaling vision with sparse mixture of experts." NeurIPS (2021).

---

> ### Author Response · Authors · 2024-11-22
> **Response to Reviewer JYvV (Part 1)**
>
> Dear reviewer, thank you for your thorough review and constructive feedback on our paper. We appreciate the time you have taken to provide valuable insights. Please find our responses below.
>
> **[C1] Time series models utilising Mixture of Experts (MoE) have been well studied since the 1990s [1] up until now [2,3,4]. As the work under review combines well-established MoE approaches with existing time series models for a specific task, i.e. forecasting, it offers a limited technical contribution to the field of time series analysis.**
>
> Thank you for the comment. We respectfully disagree with the reviewer’s perspective. In this study, the use of MoE is specifically designed to address the heterogeneity of time series during time series foundation model pretraining. This approach is novel in tackling this challenge, positioning Moirai-MoE as the first MoE-based foundation model for time series.
>
> Regarding the reference papers you provided, [1, 2, 3] are not time series foundation models, and their motivations and experimental setups differ significantly from ours. [4] refers to Time-MoE, which is a concurrent work being submitted to ICLR 2025. It is unfair to evaluate the novelty of our paper based on a concurrent work.
>
> **[C2] The related work section is incomplete, as the authors fail to discuss existing time series model that utilise MoE. Previous works stated above [1,2,3,4] and further works should be included by the authors to provide a representative overview of the current literature.**
>
> Thank you for this helpful comment. We have revised Section 2 of the paper (highlighted in red).
>
> **[C3] The methodology section of the paper is poorly elaborated.**
>
> Thank you for this helpful comment. We have revised Section 3.2 of the paper to address this point, with the changes highlighted in red.
>
> **[C4] The authors do not elaborate on the limitations of their work.**
>
> Thank you for this helpful comment. We have added this section in Appendix C of the revised paper (highlighted in red).
>
> **[C5] The authors do not provide code to support reproducibility.**
>
> Thank you for bringing up this concern. Code is available at the link: https://anonymous.4open.science/r/moirai_moe-NB88
>
> **[C6] How does the choice of pre-trained model influence the resulting cluster centroids and, consequently, the downstream performance? To this end, have the authors investigated any other pre-trained model besides Moirai?**
>
> Thank you for this insightful comment. We have conducted additional experiments, the details of which are outlined below. Specifically, we utilized the Chronos-Small checkpoint, as it matches the number of Transformer layers in Moirai-MoE-Small. Similar to Moirai-MoE, we performed inference on Chronos using our pretraining dataset, LOTSA, to extract the hidden states after the attention modules and generate cluster centroids at each layer. Subsequently, we pretrained a Moirai-MoE-Chronos variant using the token clusters derived from Chronos. The results on Monash, presented below, show that the performance of this variant is comparable to Moirai-MoE, demonstrating that our proposed gating function is capable of leveraging knowledge from other foundation models.
>
> | Variant | Pretrained Model | Inference Data for Cluster Generation| Layers | d_ff | Performance
> | -------- | ------- | ------- | -------  | ------- | -------
> Moirai-MoE-Chronos | Chronos-Small | LOTSA | 6 | 512 | 0.67
> Moirai-MoE | Moirai-Small | LOTSA | 6 | 512 | 0.65

---

> > ### Author Response · Authors · 2024-11-22
> > **Response to Reviewer JYvV (Part 2)**
> >
> > **[C7] The experiments should include existing time series models that use MoE, e.g. [3,4], to enable a fair comparison of the proposed MoE approach. Furthermore, the results should be reported over multiple seeds to ensure robustness. Well established metrics such as the mean squared error (MSE) or mean absolute error (MAE) should be reported to enable fair comparison with methods that are not included as baselines.**
> >
> > Thank you for raising this concern. The reported results have been aggregated over multiple seeds. Please note that Time-MoE is a concurrent work being submitted to ICLR 2025. While a direct comparison is not necessary (https://iclr.cc/Conferences/2025/FAQ), we have still included the evaluation results. Below, we provide additional zero-shot forecasting results for the methods of MoLE-DLinear, TTM, Timer, Moment, Time-MoE-Base, and Time-MoE-Large. We use asterisks* to mark the non-zero-shot datasets because they were used in the pretraining corpora of baselines. Additionally, we provide two average values for fair comparisons: one that averages across all datasets and another, referred to as the non-leak average, which excludes the Electricity and Solar datasets. The results confirm the superiority of Moirai-MoE over the compared baselines. We have revised Table 2 in the paper to address the issue.
> >
> > | Method | Metric | Electricity | Solar | Power | ETT1 | ETT2 | Traffic | MDENSE | Walmart | Weather | BizITObs | Avg (all) | Avg (non-leak)
> > | -------- | ------- | ------- | ------- | ------- | ------- | ------- | ------- | ------- | ------- | ------- | ------- | ------- | -------
> > | Seasonal Naive | CRPS | 0.070 | 0.512 | 0.085 | 0.515 | 0.205 | 0.257 | 0.294 | 0.151 | 0.068 | 0.262 | 1.000 | 1.000
> > |   | MASE | 0.881 | 1.203 | 0.906 | 1.778 | 1.390 | 1.137 | 1.669 | 1.236 | 0.782 | 0.986 | 1.000 | 1.000
> > | MoLE-DLinear | CRPS | 0.083 | 0.535 | 0.072 | 0.344 | 0.188 | 0.237 | 0.108 | 0.137 | 0.079 | 0.095 | 0.780 | 0.714
> > |   | MASE | 0.984 | 1.257 | 1.325 | 1.606 | 3.194 | 1.016 | 0.914 | 1.115 | 0.925 | 0.282 | 0.938 | 0.906
> > | TTM | CRPS | 0.075 | 0.534* | 0.059 | 0.417 | 0.122 | 0.210 | 0.150 | 0.192 | 0.055 | 0.102 | 0.758 | 0.697
> > |    | MASE | 0.802 | 1.255* | 0.898 | 1.934 | 1.547 | 0.901 | 1.195 | 1.477 | 0.506 | 0.308 | 0.831 | 0.798
> > | Timer | CRPS | 0.084 | 0.573 | 0.066 | 0.345 | 0.135 | 0.182 | 0.152 | 0.151 | 0.092 | 0.120 | 0.797 | 0.726
> > |    | MASE | 0.967 | 1.344 | 1.006 | 1.697 | 1.754 | 0.770 | 1.196 | 1.219 | 0.655 | 0.376 | 0.871 | 0.820
> > | Moment | CRPS | 0.354 | 1.332 | 0.151 | 0.401 | 0.277 | 0.612 | 0.157 | 0.154 | 0.105 | 0.313 | 1.502 | 1.205
> > |    | MASE | 3.167 | 3.139 | 2.244 | 2.243 | 4.100 | 2.617 | 1.277 | 1.245 | 1.053 | 0.913 | 1.691 | 1.457
> > | Time-MoE-Base | CRPS | 0.051* | 0.230* | 0.044 | 0.392 | 0.125 | 0.152 | 0.099 | 0.100 | 0.070 | 0.112 | 0.583 | 0.586
> > |    | MASE | 0.587* | 0.535* | 0.800 | 1.823 | 1.672 | 0.672 | 0.846 | 0.833 | 0.558 | 0.343 | 0.662 | 0.695
> > | Time-MoE-Large | CRPS | 0.051* | 0.294* | 0.045 | 0.386 | 0.131 | 0.172 | 0.090 | 0.097 | 0.058 | 0.111 | 0.589 | 0.576
> >  |   | MASE | 0.581* | 0.689* | 0.790 | 1.773 | 1.878 | 0.762 | 0.759 | 0.817 | 0.524 | 0.337 | 0.678 | 0.695
> > | Moirai-MoE-Small | CRPS | 0.046 | 0.429 | 0.036 | 0.288 | 0.093 | 0.108 | 0.071 | 0.090 | 0.056 | 0.081 | 0.497 | 0.450
> > |    | MASE | 0.719 | 1.222 | 0.737 | 1.750 | 1.248 | 0.563 | 0.746 | 0.927 | 0.476 | 0.298 | 0.670 | 0.620
> > Moirai-MoE-Base | CRPS | 0.041 | 0.382 | 0.034 | 0.296 | 0.091 | 0.100 | 0.071 | 0.088 | 0.057 | 0.079 | **0.478** | **0.439**
> > |    | MASE | 0.638 | 1.161 | 0.725 | 1.748 | 1.247 | 0.510 | 0.721 | 0.918 | 0.509 | 0.290 | **0.651** | **0.611**

---

> > > ### Author Response · Authors · 2024-11-22
> > > **Response to Reviewer JYvV (Part 3)**
> > >
> > > Regarding metrics, we have used the MAE metric in our evaluation on Monash. To address your concerns, we report the MSE and MAE for zero-shot predictions below, and also in Appendix A, Table 8 of the revised paper (highlighted in red). We bold both the best and second best results here. Based on these, Moirai-MoE achieves the best performance when considering the non-leak average values.
> > >
> > > | Method | Metric | Electricity | Solar | Power | ETT1 | ETT2 | Traffic | MDENSE | Walmart | Weather | BizITObs | Avg (all) | Avg (non-leak)
> > > | -------- | ------- | ------- | ------- | ------- | ------- | ------- | ------- | ------- | ------- | ------- | ------- | ------- | -------
> > > Seasonal Naive | MSE | 1299429.16 | 1293.24 | 1798196.83 | 57976.63 | 122878.95 | 203.32 | 39929.67 | 32876026.66 | 2197.23 | 174.31 | 1.000 | 1.000
> > > | | MAE | 166.20 | 15.77 | 492.60 | 154.98 | 211.56 | 8.72 | 118.38 | 2637.43 | 10.96 | 9.69 | 1.000 | 1.000
> > > iTransformer | MSE | 1264494.38 | 1183.57 | 968959.56 | 55320.57 | 178757.02 | 41.77 | 9905.39 | 10922819.00 | 1885.01 | 20.55 | 0.508 | 0.435
> > > | | MAE | 165.89 | 17.61 | 399.09 | 170.83 | 279.21 | 4.85 | 51.06 | 1560.68 | 10.65 | 2.66 | 0.741 | 0.678
> > > MoLE-DLinear | MSE | 1901617.97 | 1098.56 | 1071490.46 | 39026.37 | 195287.19 | 153.71 | 13016.78 | 26832049.08 | 1649.90 | 21.57 | 0.656 | 0.575
> > > | | MAE | 197.06 | 16.47 | 420.67 | 130.79 | 328.28 | 8.48 | 62.43 | 2395.50 | 12.81 | 2.75 | 0.857 | 0.803
> > > TimesFM | MSE | 1378828.95* | 1061.70 | 384815.80 | 42789.02 | 169714.41 | 106.01 | 10194.73 | 9494507.86 | 1317.09 | 23.23 | 0.475 | 0.401
> > > | | MAE | 137.57* | 18.07 | 277.94 | 138.42 | 245.61 | 5.75 | 49.78 | 1484.68 | 7.94 | 2.89 | 0.672 | **0.612**
> > > TTM | MSE | 2432897.66 | 884.33* | 647289.67 | 56256.46 | 116203.30 | 114.79 | 18425.62 | 39297380.00 | 1122.55 | 23.41 | 0.625 | 0.538
> > > | | MAE | 179.56 | 16.46* | 341.96 | 158.85 | 213.61 | 7.53 | 86.44 | 3360.79 | 8.88 | 2.97 | 0.833 | 0.784
> > > Timer | MSE | 2205084.30 | 962.26 | 687600.25 | 39235.36 | 129063.67 | 75.23 | 19875.60 | 29410540.00 | 1873.68 | 27.21 | 0.613 | 0.527
> > > | | MAE | 200.62 | 17.57 | 370.53 | 131.31 | 235.27 | 6.42 | 87.72 | 2646.92 | 13.65 | 3.50 | 0.865 | 0.804
> > > Moment | MSE | 44303358.90 | 2876.47 | 3272382.39 | 46075.47 | 411967.28 | 601.62 | 19506.54 | 29046437.85 | 1804.48 | 129.26 | 1.760 | 1.180
> > > | | MAE | 843.45 | 41.02 | 873.48 | 152.56 | 484.86 | 21.87 | 90.51 | 2690.84 | 16.89 | 9.11 | 1.650 | 1.355
> > > Chronos-Small | MSE | 1251170.49* | 1405.10* | 418195.72 | 60157.02 | 112472.02 | 100.62 | 15377.29 | 14697271.28 | 3945.04 | 23.89 | 0.587 | 0.511
> > > | | MAE | 126.25* | 15.79* | 275.11 | 161.23 | 207.11 | 5.28 | 59.26 | 1693.33 | 16.90 | 2.94 | 0.724 | 0.691
> > > Chronos-Base | MSE | 1147348.35* | 1062.73* | 400709.37 | 66320.26 | 107178.21 | 80.48 | 12770.66 | 15813384.14 | 1720.53 | 22.78 | 0.501 | 0.439
> > > | | MAE | 121.69* | 13.18* | 285.79 | 169.60 | 194.70 | 4.69 | 51.58 | 1706.11 | 10.28 | 2.82 | 0.656 | 0.628
> > > Chronos-Large | MSE | 1073679.39* | 1017.98* | 362386.33 | 73974.48 | 106362.90 | 98.20 | 13625.07 | 12339319.84 | 1874.83 | 23.61 | 0.503 | 0.447
> > > | | MAE | 121.06* | 12.86* | 277.64 | 177.68 | 191.07 | 5.07 | 53.61 | 1560.11 | 11.30 | 2.89 | 0.664 | 0.639
> > > Moirai-Small | MSE | 4015423.50 | 1429.82 | 757613.06 | 39481.46 | 118636.33 | 146.24 | 11041.41 | 19886286.00 | 1932.16 | 22.48 | 0.647 | 0.498
> > > | | MAE | 219.02 | 19.19 | 358.01 | 133.82 | 209.68 | 8.71 | 58.25 | 2112.07 | 10.23 | 2.90 | 0.802 | 0.715
> > > Moirai-Base | MSE | 1734656.25 | 1105.95 | 477193.47 | 51793.64 | 113074.23 | 44.60 | 17724.71 | 18981036.00 | 1196.21 | 22.44 | 0.500 | 0.414
> > > | | MAE | 164.94 | 16.97 | 293.74 | 149.15 | 202.89 | 4.72 | 79.41 | 2046.22 | 7.73 | 2.81 | 0.713 | 0.650
> > > Moirai-Large | MSE | 1229872.00 | 997.13 | 340307.44 | 44752.48 | 106513.38 | 101.17 | 14874.89 | 21274060.00 | 1914.39 | 21.79 | 0.511 | 0.449
> > > | | MAE | 150.66 | 16.25 | 262.70 | 142.21 | 204.72 | 5.93 | 69.73 | 2110.73 | 10.10 | 2.77 | 0.720 | 0.669
> > > Time-MoE-Base | MSE | 1158323.38* | 176.27* | 315704.91 | 50267.22 | 114374.42 | 89.87 | 11303.31 | 13934856.92 | 1371.87 | 28.51 | **0.395** | 0.408
> > > | | MAE | 120.52* | 7.07* | 254.28 | 149.21 | 218.55 | 5.70 | 57.43 | 1742.96 | 11.35 | 3.26 | **0.644** | 0.663
> > > Time-MoE-Large | MSE | 1203643.75* | 194.84* | 350989.67 | 47389.70 | 121112.59 | 99.13 | 9585.73 | 12876789.32 | 1264.26 | 27.34 | **0.394** | 0.400
> > > | | MAE | 120.53* | 9.06* | 262.48 | 147.11 | 229.67 | 6.45 | 52.10 | 1687.08 | 9.32 | 3.24 | 0.650 | 0.652
> > > Moirai-MoE-Small | MSE | 930140.63 | 1113.50 | 360995.59 | 45412.81 | 114609.09 | 53.05 | 9426.45 | 18025986.00 | 1944.27 | 23.45 | 0.453 | **0.395**
> > > | | MAE | 138.03 | 16.05 | 260.82 | 141.08 | 194.63 | 4.78 | 50.09 | 1955.77 | 10.08 | 2.89 | 0.668 | 0.617
> > > Moirai-MoE-Base | MSE | 907276.31 | 1047.63 | 311227.06 | 48487.21 | 107284.42 | 45.83 | 9740.51 | 17094764.00 | 1954.24 | 22.54 | 0.434 | **0.378**
> > > | | MAE | 122.27 | 15.24 | 251.10 | 145.50 | 191.47 | 4.33 | 49.73 | 1919.31 | 10.31 | 2.80 | **0.646** | **0.605**

---

> > > > ### Author Response · Authors · 2024-11-22
> > > > **Response to Reviewer JYvV (Part 4)**
> > > >
> > > > **[C8] In Table 1, rather than presenting the number of activated paramters, it would be more insightful to show how the downstream performance of Moirai-MoE evolves during training, e.g. with increasing FLOPs and GPU hours, compared to Moirai. Does Moirai-MoE match or even outperform Moirai with less compute? If so, at what point during training do the advantages of the MoE layers become evident?**
> > > >
> > > > Thank you for this insightful comment. In Appendix B.1, Figure 8, we provide a comparison between Moirai-Small and Moirai-MoE-Small in terms of training steps. The results demonstrate that Moirai-MoE outperforms Moirai from the very first evaluation point -- 25k steps. Furthermore, Moirai-MoE at 25k steps achieves better performance than Moirai at 125k steps. This figure highlights the clear advantages of Moirai-MoE in terms of both model performance and reduced training steps.
> > > >
> > > > **[C9] The trends depicted in Figure 4 indicate that it might be worth increasing the training steps (larger than 100k) and the number of experts (larger than 32). To this end, could the authors extend their analysis to see whether downstream performance can be further improved?**
> > > >
> > > > Thank you for this helpful comment. We have conducted a further investigation into the impact of training steps and the total number of experts. As shown in Figure 4, increasing the training steps to 125k and the number of experts to 64 does not yield additional performance improvements.
> > > >
> > > > **[C10] In Figure 5, it seems that experts of the Covid Daily Deaths are a subset of the experts allocated for the Traffic Hourly data. Could the authors explain whether this is based on data similarity or any other phenomenon?**
> > > >
> > > > Thank you for your thoughtful feedback. We agree with the reviewer that, in terms of specific expert selection, Covid Daily is a subset of Traffic Hourly. However, their ratios for selecting experts are significantly different. This is because these two datasets exhibit very distinct patterns (see Figure 13 and Figure 14), leading to divergent expert allocation distributions.
> > > >
> > > >
> > > > **[C11] In Figure 6, the expert allocation appears very similar across frequencies and is notably sparse, suggesting that some experts might be redundant. How does this align with Figure 4, which indicates that downstream performance improves as the number of experts increases?**
> > > >
> > > > Thank you for this insightful comment. We agree with the reviewer that some experts in Moirai-MoE are rarely selected during inference on the datasets from Monash. Pruning these underutilized experts to improve inference efficiency is an interesting direction and is left for future work.
> > > >
> > > > However, as shown in Figure 4 (right part), we observe that downstream forecasting performance consistently improves with an increasing number of experts. This suggests that scaling up the number of experts during training is necessary, while we can certainly prune the experts or perform other model quantization methods to accelerate model inference.
> > > >
> > > > **[C12] Furthermore, is it necessary to replace all FFNs with MoE layers? Existing work on MoE in natural language processing [5] and computer vision [6] show that replacing only certain layers has little impact on downstream performance, while saving computation time due to less communication overhead.**
> > > >
> > > > Thank you for the insightful comment. This is an open question, and there are many options for replacing FFN layers with MoE layers. According to a recent survey on MoE-based LLMs [1], replacing all FFNs with MoE layers has become a common practice in recent and well-known LLMs, such as Mixtral 8x7B and Qwen1.5-MoE. Another popular choice is to replace half of the FFN layers with MoE layers.
> > > >
> > > > To address your concern, we have conducted experiments where one out of every two FFN layers was replaced with MoE layers. Compared to replacing all FFN layers, this "half" setting reduces training time by 31%. The downstream performance on the Monash benchmark is presented below, showing that replacing half of the FFN layers results in a 5% performance drop. This demonstrates a trade-off between performance and training cost.
> > > >
> > > > | Variant | MoE Layers | Total Layers | Activated Params | Total Params | Performance | Pretraining Time
> > > > | -------- | ------- | ------- | -------  | ------- | ------- | -------
> > > > Moirai-MoE-Small-Half | 3 | 6 | 11M | 64M | 0.68 | 6.53h
> > > > Moirai-MoE-Small | 6 | 6 | 11M | 117M | 0.65 | 9.49h
> > > >
> > > > [1] A Survey on Mixture of Experts. https://arxiv.org/pdf/2407.06204.

---

> > > > > ### Author Response · Authors · 2024-11-22
> > > > > **Response to Reviewer JYvV (Part 5)**
> > > > >
> > > > > **[C13] Have the authors analysed what the experts learn? Do some experts focus on local patterns while others focus on global patterns? Do some experts analyse low-frequency components while others analyse high-frequency components? The authors attempt to explain expert allocation with reference to Figure 7, however, the figure is poorly structured and does not support clear conclusions.**
> > > > >
> > > > > Thank you for asking this insightful question. To address the concern, we have added experiments to investigates the relationship between raw time series observations and their corresponding expert allocations. In Appendix B.5, Figure 11, the upper subfigure presents a Traffic Hourly time series sequence with a length of 512. For enhanced visualization, the sequence is segmented using vertical dashed lines, each spanning 16 steps, which is equal to the length of a single time series token. The lower subfigure illustrates the expert allocations at shallow layers for 32 tokens derived from the 512 observations. The yellow straight line represents the specific experts selected by the token at each position. The alignment of subfigures facilitates an intuitive comparison between the time series trends and the associated expert selections.
> > > > >
> > > > > The figure includes red square boxes to highlight time series segments exhibiting a downward trend followed by a slight upward pattern. These segments consistently correspond to the activation of two specific experts, as shown in the lower subfigure. This observation suggests that Moirai-MoE effectively captures time-based structures and demonstrates model specialization at the token level.
> > > > >
> > > > > **[C14] The authors provide an efficiency analysis focused on the inference cost of predicting a single token, i.e. a forecasting horizon of 1 token. While this does not allow for a fair evaluation of real-world applications, the authors might rather investigate how inference cost scales with an increasing forecasting horizon.**
> > > > >
> > > > > Thanks for raising this concern. The main goal of our efficiency analysis is to validate whether the dense model and the MoE model with similar activated parameters achieve comparable speeds, as they theoretically should. However, due to differences in the inference algorithms of Moirai and Moirai-MoE, we conducted experiments using a single-token prediction setting to mitigate the gap.
> > > > >
> > > > > To investigate the scaling of inference cost, we have expanded this analysis to include predictions with more tokens. We set 1 token to correspond to 16 time steps. The below table presents the spent time results (in seconds). For each predicted token evaluated here, we select a subset of datasets from the Monash benchmark such that the total number of test samples of all subsets is approximately the same.
> > > > >
> > > > > When predicting a single token, Moirai and Moirai-MoE, with comparable activated parameters, demonstrate similar inference times. As the number of tokens increases to 11, Moirai's inference time grows by 3 to 5 times. This increase is attributed to masked tokens in Moirai also being involved in the attention computation. For the autoregressive models, Moirai-MoE and Chronos, Moirai-MoE is significantly faster. This speed advantage is due to Moirai-MoE predicting using patches of size 16, whereas Chronos operates with a patch size of 1, which considerably impacts its inference efficiency.
> > > > >
> > > > > | #Predicted Tokens | #Test Samples | Chronos-Small | Chronos-Base  | Chronos-Large | Moirai-Small | Moirai-Base | Moirai-Large | Moirai-MoE-Small | Moirai-MoE-Base
> > > > > | -------- | ------- | ------- | -------  | ------- | ------- | ------- | -------  | -------  | -------
> > > > > | | | (46M)  | (200M)  | (710M)  | (14M)  | (91M)  | (311M)  | (11M/117M)  | (86M/935M)
> > > > > 1 Token | 1,194 | 5.81 | 7.33 | 13.12 | 3.32 | 4.02 | 5.60 | 3.13 | 3.73
> > > > > 2 Tokens | 1,212 | 13.06 | 21.87 | 54.99 | 4.26 | 4.87 | 6.03 | 6.94 | 18.93
> > > > > 11 Tokens | 1,183 | 153.87 | 437.42 | 1125.98 | 16.35 | 17.71 | 18.56 | 51.20 | 197.17
> > > > >
> > > > > **[C15] Regarding Figure 3 and Table 2, do the authors plan to provide full results for a fair comparison?**
> > > > >
> > > > > Thank you for asking this. The complete results for Figure 3 are already provided in Appendix A, Table 6. For Table 2, we have included additional baselines and metrics. The results are shown in Table 2 and Table 8.
> > > > >
> > > > > We are very grateful for your insightful comments, which have certainly given us an opportunity to fine-tune our messaging. Your expert knowledge is helping us to strengthen the manuscript significantly. Please let us know if you have any other comments/questions.

---

> > > > > > ### Author Response · Authors · 2024-11-25
> > > > > > **Kindly Request for Reviewer's Feedback**
> > > > > >
> > > > > > Dear reviewer JYvV,
> > > > > >
> > > > > > Thank you for taking the time and effort in providing a valuable review of our work. As the discussion period is coming to a close, we hope that you have had the chance to review our rebuttal. We believe that our rebuttal has provided additional clarity of our work. If our response has addressed your concerns, we hope that you could kindly leave an update to reflect this, and are more than willing to engage in further discussion if needed.

---

> > > > > > > ### Author Response · Authors · 2024-11-28
> > > > > > > **Kindly Request for Reviewer's Feedback**
> > > > > > >
> > > > > > > Dear Reviewer JYvV,
> > > > > > >
> > > > > > > Thank you once again for your insightful feedback on our paper. As the deadline for submitting the revised PDF approaches, we wanted to check whether our earlier responses sufficiently addressed your 15 comments. If there are any additional points you’d like us to clarify or elaborate on, please let us know.

---

> ### Author Response · Authors · 2024-12-01
> **Request of Reviewer's Attention and Feedback**
>
> Dear reviewer JYvV,
>
> Thanks for your valuable review, which has inspired us to improve our paper further. May we know if our response addresses your main concerns? If so, we kindly ask for your reconsideration of the score. We look forward to hearing your thoughts on our revisions.

---

> > ### Author Response · Authors · 2024-12-02
> > **Last Reminder to Reviewer**
> >
> > Dear reviewer JYvV,
> >
> > Thank you once again for your valuable feedback on our paper. As we enter the last 24 hours of the reviewer response period, we would like to provide a brief summary of the key updates and clarifications made in our rebuttal:
> >
> > **Experiments:**
> > 1. Added 6 additional baselines, including MoLE-DLinear, TTM, Timer, Moment, Time-MoE-Base, and Time-MoE-Large.
> > 2. Added results for MSE and MAE metrics across 17 baselines on 10 datasets.
> > 3. Investigated the impact of pretrained model choices on performance.
> > 4. Examined the effects of replacing half of the FFN layers with MoE layers.
> > 5. Analyzed how inference costs scale with increasing forecasting lengths.
> > 6. Added a detailed figure illustrating what the experts learned and how they capture input patterns.
> > 7. Compared training curves between Moirai and Moirai-MoE.
> > 8. Reported performance when increasing training steps to 125k and the number of experts to 64.
> >
> > **Arguments:**
> > 1. Clarified the novelty of Moirai-MoE and compared it with prior MoE time series works in Section 2.
> > 2. Enhanced the methodology description in Section 3.2.
> > 3. Added a limitation section.
> > 4. Open-sourced the evaluation code to ensure reproducibility.
> > 5. Clarified the necessity of using 32 experts in the model.
> >
> > We hope these responses address your concerns and provide the necessary clarity. As the rebuttal period concludes, we hope that you have had the chance to review our rebuttal. Thank you again for your constructive feedback.

---

> ### Comment · Reviewer_JYvV · 2024-12-03
> **Reviewer Response**
>
> Thank you very much for the response and the revised manuscript. I have carefully read both the rebuttal and the updated version of the study.
>
> The authors have conducted an interesting study on whether Mixture of Experts (MoE) can further improve time series foundation models, such as Moirai [1].
>
> However, I still find the contribution of this work to be limited, given that MoE approaches have been well studied in the literature, including the field of time series analysis [2][3][4][5]. Additionally, the combination of two well established components, i.e. Moirai [1] and MoE [2][6][7][8][9], are not well studied in this manuscript. The methodology section is still very poor, particularly when compared to other impactful MoE studies [10][11]. The experimental section still leaves concerns on whether MoE approaches have been effectively combined with the Moirai [1] model. For instance, Figures 5 and 6 indicate that crucial components, such as the number of experts, were not carefully investigated. The reliance on pre-trained model-based clustering raises questions about whether the study has sufficiently explored alternatives. The rebuttal mentions that other pre-trained models [12] might also be applicable - or even more suitable - but these possibilities were not investigated. Visualisations such as in Figure 7 still fail to convey a clear and intuitive message on the expert allocation, and thus do not allow for any insightful conclusion.
>
> Given these points, I cannot recommend accepting the study at this time, and thus leave my scores unchanged. However, I strongly encourage the authors to further improve the quality of the study, as the conceptual idea is interesting and has potential for significant impact with further refinement.
>
> ---
> [1] Woo et al. "Unified Training of Universal Time Series Forecasting Transformers." ICML (2024).
>
> [2] Zeevi et al. "Time series prediction using mixtures of experts." NeurIPS (1996).
>
> [3] Yuksel et al. "Twenty years of mixture of experts." IEEE transactions on neural networks and learning systems (2012).
>
> [4] Ni et al. "Mixture-of-Linear-Experts for Long-term Time Series Forecasting." AISTATS (2024).
>
> [5] Shi et al. "Time-MoE: Billion-Scale Time Series Foundation Models with Mixture of Experts." arXiv (2024).
>
> [6] Miller et al. "A mixture of experts classifier with learning based on both labelled and unlabelled data." NeurIPS (1996).
>
> [7] Jacobs et al. "Adaptive mixtures of local experts." Neural computation (1991).
>
> [8] Xu et al. "An alternative model for mixtures of experts." NeurIPS (1994).
>
> [9] Hu et  al. "A patient-adaptable ECG beat classifier using a mixture of experts approach." IEEE transactions on biomedical engineering (1997).
>
> [10] Lepikhin et al. "GShard: Scaling giant models with conditional computation and automatic sharding." ICLR (2021).
>
> [11] Riquelme et al. "Scaling vision with sparse mixture of experts." NeurIPS (2021).
>
> [12] Ansari et al. "Chronos: Learning the language of time series." arXiv (2024).

---

> ### Author Response · Authors · 2024-12-03
> **Response to Reviewer JYvV (Part 6)**
>
> Dear reviewer, thank you much for your reply. Please find our responses below.
>
> **[C16] However, I still find the contribution of this work to be limited, given that MoE approaches have been well studied in the literature, including the field of time series analysis [2][3][4][5].**
>
> Thanks for the comment. We respectfully disagree with this point. To clarify, we would like to restate our motivation for employing MoE. We target the challenges raising from unified time series training for foundation models. As the large and diverse time series pre-training data is highly heterogeneous, existing approaches introduce some level of model specialization to account for such heterogeneity. We identify their drawbacks and propose to automatically achieve model specialization at the token level, and we utilize the MoE concept to do this. This is a novel aspect for tackling the challenge of unified time series training for foundation models.
>
> Regarding the references provided by the reviewer:
>
> [2] focuses on theoretical analyses, demonstrating that a mixture of linear autoregressive models serves as a universal approximator, akin to neural networks.
>
> [3] is a survey paper.
>
> [4] introduces multiple linear-centric models alongside a router that weighs and combines their outputs.
>
> [5] is a concurrent work that uses standard MoE techniques to scale up time series foundation models. The same motivation as MoE usage in large language models.
>
> As we can see, although titles contain "mixture of experts", none of these works share the same motivation for utilizing the MoE concept as we do.
>
> **[C17] Additionally, the combination of two well established components, i.e. Moirai [1] and MoE [2][6][7][8][9], are not well studied in this manuscript.**
>
> Thank you for the comment. We respectfully disagree with the reviewer's request to evaluate the MoE methods listed in [2][6][7][8][9] with our model.
>
> Our work targets the challenges raising from unified time series training. We utilize the MoE concept to achieve token-level model specialization and thus to account for the highly heterogeneous nature of time series data. This is actually not related to the specific designs of MoE.
>
> Additionally, we have explored three MoE variants in our work: linear projection, linear projection with load balancing loss, and token clustering. Experimental results demonstrate that all these variants outperform prior foundation models.
>
> Lastly, thanks for listing these MoE works. We will include and discuss the papers you suggested in the related work section of our next updated version. However, due to the last-minute of your reply, we are unable to update the current PDF.
>
> **[C18] The methodology section is still very poor, particularly when compared to other impactful MoE studies [10][11].**
>
> The initial review lists six specific points where our writing of the method section could be improved. We have carefully addressed these points in our revised submission. However, the reviewer continues to find the method section "very poor." Could you please specify which aspects remain unsatisfactory? This would help us better understand your concerns and address them effectively in the updated version of our paper.
>
> Regarding the references ([10] and [11]) provided by the reviewer:
>
> [10] dedicates significant content to detailing the implementation and parallel training of MoE on hardware such as TPUs. While this is a valuable discussion, it is not the primary focus of our paper.
>
> [11] is a pioneering study on using MoE in the vision domain. In its method section, the authors first provide an overview of MoE, then explain how it can be applied in the vision domain, and finally describe the design of the gating function. This structure aligns closely with our approach in Section 3.2 of our paper.
>
> Could the reviewer clarify the specific differences between the writing style or organization in reference [11] and our paper? This would enable us to address your concerns more thoroughly.

---

> ### Author Response · Authors · 2024-12-03
> **Response to Reviewer JYvV (Part 7)**
>
> **[C19] The experimental section still leaves concerns on whether MoE approaches have been effectively combined with the Moirai [1] model. For instance, Figures 5 and 6 indicate that crucial components, such as the number of experts, were not carefully investigated.**
>
> First, we respectfully disagree with the point "experimental section leaves concerns that whether MoE have been effectively combine with Moirai". In fact, the experimental results demonstrate the significant effectiveness of Moirai-MoE. In zero-shot forecasting, Moirai-MoE-Small achieves a 3%–14% improvement in CRPS and an 8%–16% improvement in MASE compared to all sizes of Moirai. These improvements are particularly remarkable given that Moirai-MoE-Small has only 11 million activated parameters, which is 28 times fewer than Moirai-Large.
>
> Second, Figures 5 and 6 reveal that some experts in the deeper layers of Moirai-MoE are rarely selected. Rather than being a flaw, we think this is an intriguing finding that offers insights into the inner workings of time series foundation models. Our findings suggest that denoising processes occur progressively throughout the model. This observation is consistent with conclusions from studies [1] and [2], which indicate that, as layer depth increases, certain token vectors become redundant and are projected into the low-dimensional space defined by the top eigenvectors of the input patterns.
>
> [1] Exploring Representations and Interventions in Time Series Foundation Models.
>
> [2] One Fits All:Power General Time Series Analysis by Pretrained LM.
>
> **[C20] The reliance on pre-trained model-based clustering raises questions about whether the study has sufficiently explored alternatives. The rebuttal mentions that other pre-trained models [12] might also be applicable - or even more suitable - but these possibilities were not investigated.**
>
> Thanks for the comment. This concern relates to [C6]. To address it, we conducted additional experiments using the pre-trained Chronos-Small checkpoint, which matches the number of Transformer layers in Moirai-MoE-Small. The results on the Monash benchmark (the lower the better), presented below, show that the performance of this variant is comparable to Moirai-MoE, demonstrating that our proposed gating function is capable of leveraging knowledge from other foundation models.
>
> | Variant | Pretrained Model | Inference Data for Cluster Generation| Layers | d_ff | Performance (the lower the better)
> | -------- | ------- | ------- | -------  | ------- | -------
> Moirai-MoE-Chronos | Chronos-Small | LOTSA | 6 | 512 | 0.67
> Moirai-MoE | Moirai-Small | LOTSA | 6 | 512 | 0.65
>
> We would like to clarify that we have never claimed the Chronos checkpoint to be more suitable.
>
> We note that the reviewer posted this comment 4.5 hours before the deadline, leaving us insufficient time to add more experiments. Additionally, the suggestion to "sufficiently explore alternatives" lacks clarity. Could the reviewer provide more specific guidance or examples of the alternatives you believe should be explored? This would help us address your concern more effectively.
>
>
> **[C21] Visualisations such as in Figure 7 still fail to convey a clear and intuitive message on the expert allocation, and thus do not allow for any insightful conclusion.**
>
> Thanks for the comment. The complaint about Figure 7 aligns with the comment raised in [C13]. To address [C13], we have added experiments to investigates the relationship between raw time series observations and their corresponding expert allocations. In Appendix B.5, Figure 11, the upper subfigure presents a Traffic Hourly time series sequence with a length of 512. For enhanced visualization, the sequence is segmented using vertical dashed lines, each spanning 16 steps, which is equal to the length of a single time series token. The lower subfigure illustrates the expert allocations at shallow layers for 32 tokens derived from the 512 observations. The yellow straight line represents the specific experts selected by the token at each position. The alignment of subfigures facilitates an intuitive comparison between the time series trends and the associated expert selections.
>
> The figure includes red square boxes to highlight time series segments exhibiting a downward trend followed by a slight upward pattern. These segments consistently correspond to the activation of two specific experts, as shown in the lower subfigure. This observation suggests that Moirai-MoE effectively captures time-based structures and demonstrates model specialization at the token level.

---

### Official Review · Reviewer_xfCg · 2024-10-31

**Soundness:** 2
**Presentation:** 3
**Contribution:** 2
**Rating:** 6
**Confidence:** 5

**Summary:**

In response to the variability of temporal patterns in pre-trained time series datasets, this paper proposes the MOIRAI-MOE. It leverages token embeddings from existing time series pre-trained models to derive cluster centers, resulting in a more effective gating function. Extensive experiments show that MOIRAI-MOE achieves enhanced predictive accuracy and overall superior performance.

**Strengths:**

- The article proposes the application of MOE to time series foundational models, achieving token-level specialization to enable the model to handle temporal data with significant heterogeneity in temporal patterns.
- Through extensive experiments, the paper demonstrates the effectiveness of this method, achieving impressive results with high inference efficiency.
- The paper provides insights through experimental analysis, such as how heterogeneous experts reflect the periodic pattern information of the time series data.

**Weaknesses:**

- **Methodology Limitations:** First, MOE is a well-known technique in LLMs, so its direct application results in limited novelty for the article. Furthermore, the gating functions used in the MOE depend on the temporal embeddings from pre-trained models, which constrains the effectiveness of this approach.
- **Experimental Setup:**
    - The article lacks a detailed description of the experimental setup, including input and output lengths, which significantly impact the results. Additionally, it does not use established evaluation metrics (e.g., MAE, MSE) that correspond to previous mainstream studies. This omission makes it difficult for reviewers to intuitively compare the results with earlier time series prediction works, thereby reducing the credibility of the model's effectiveness.
    - Furthermore, the paper does not include recent significant works in time series foundational models, such as Timer[1], Moment[2], TTM[3], and Time-MOE[4].
- In Appendix Section B.1, the ablation study on patch size reveals that it significantly affects the model's performance. This raises the question of whether multi-patch projection is indeed an effective strategy, suggesting that MOE may not adequately address this issue, which contradicts the article's motivation.

**References**

[1] Liu, Y., Zhang, H., Li, C., Huang, X., Wang, J., & Long, M. (2024). Timer: Transformers for time series analysis at scale. arXiv preprint arXiv:2402.02368.

[2] Goswami, M., Szafer, K., Choudhry, A., Cai, Y., Li, S., & Dubrawski, A. (2024). Moment: A family of open time-series foundation models. arXiv preprint arXiv:2402.03885.

[3] Ekambaram, V., Jati, A., Nguyen, N. H., Dayama, P., Reddy, C., Gifford, W. M., & Kalagnanam, J. (2024). TTMs: Fast Multi-level Tiny Time Mixers for Improved Zero-shot and Few-shot Forecasting of Multivariate Time Series. arXiv preprint arXiv:2401.03955.

[4] Shi, X., Wang, S., Nie, Y., Li, D., Ye, Z., Wen, Q., & Jin, M. (2024). Time-MoE: Billion-Scale Time Series Foundation Models with Mixture of Experts. arXiv preprint arXiv:2409.16040.

**Questions:**

- Could the authors provide detailed experimental setups, including the input and prediction lengths for each model across the datasets, as well as a comparison table with previous mainstream time series prediction works? Specifically, the prediction lengths should be (96, 192, 336, 720) to facilitate intuitive comparisons for reviewers.
- The authors need to include recent works on time series foundational models to better substantiate the effectiveness of your model.
- Could the authors explain why patch size still significantly affects the model, and whether multi-patch projection remains necessary under this phenomenon?
- Could the authors provide insight into whether the token clusters generated by different time series foundational models affect the performance of MOIRAI-MOE?

---

> ### Author Response · Authors · 2024-11-22
> **Response to Reviewer xfCg (Part 1)**
>
> Dear reviewer, we would like to sincerely thank you for the time and effort put into reviewing our submission. We are grateful for your acknowledgement of the effectiveness of our approach. Please find our responses below.
>
> **[C1] MOE is a well-known technique in LLMs. Its direct application results in limited novelty for the article. Furthermore, the gating functions used in the MOE depend on the temporal embeddings from pre-trained models, which constrains the effectiveness of this approach.**
>
> Thank you for the comment. We respectfully disagree with the reviewer’s perspective. In this study, the use of MoE is specifically designed to address the heterogeneity of time series during time series foundation model pretraining. This represents a novel approach to tackling this challenge, making Moirai-MoE the first MoE-based foundation model for time series.
>
> Regarding the proposed gating function, we argue that leveraging knowledge from a readily available pretrained model is not a weakness. On the contrary, it is a promising direction for MoE-based LLMs pretraining. For instance, notable models like Qwen1.5-MoE [1] initialize most of their parameters from its corresponding dense model Qwen1.5-7B before training the MoE model. Similarly, Nemotron-4-MoE [2] from NVIDIA presents a comparable approach.
>
> Furthermore, we have explored various gating mechanisms in this study. The table below provides a performance comparison on the Monash benchmark. As shown, the linear-projection-based MoE already achieves competitive performance, while the token-cluster-based MoE further enhances it. This demonstrates that the effectiveness of Moirai-MoE arises from the use of MoE itself, and it is not constrained by the utilization of pretrain models' knowledge.
>
> | Method | TimesFM | Chronos-Small | Chronos-Base | Chronos-Large | Moirai-Small | Moirai-Base | Moirai-Large | Moirai-MoE-Small-Linear | Moirai-MoE-Small-Clusters
> | -------- | ------- | ------- | ------- | ------- | ------- | ------- | ------- | ------- | -------
> | Monash | 0.77 | 0.67 | 0.71 | 0.72 | 0.78 | 0.71 | 0.70 | 0.67  | 0.65
>
> [1] Qwen1.5-MoE: Matching 7B Model Performance with 1/3 Activated Parameters.
>
> [2] Upcycling Large Language Models into Mixture of Experts.

---

> > ### Author Response · Authors · 2024-11-22
> > **Response to Reviewer xfCg (Part 2)**
> >
> > **[C2] The paper does not include recent significant works in time series foundational models, such as TTM, Timer, Moment, and Time-MOE.**
> >
> > Thank you for raising this concern. Please note that Time-MoE is a concurrent work being submitted to ICLR 2025. While a direct comparison is not necessary (https://iclr.cc/Conferences/2025/FAQ), we have still included the evaluation results.
> >
> > Below, we provide zero-shot forecasting results for the methods of MoLE-DLinear, TTM, Timer, Moment, Time-MoE-Base, and Time-MoE-Large. We have included these results in Table 2 of the revised paper. Please note that we use asterisks* to mark the non-zero-shot datasets because they were used in the pretraining corpora of baselines. Additionally, we provide two average values for fair comparisons: one that averages across all datasets, and another, referred to as the non-leak average, which excludes the Electricity and Solar datasets. The results demonstrate the superiority of Moirai-MoE over the added baselines.
> >
> > | Method | Metric | Electricity | Solar | Power | ETT1 | ETT2 | Traffic | MDENSE | Walmart | Weather | BizITObs | Avg (all) | Avg (non-leak)
> > | -------- | ------- | ------- | ------- | ------- | ------- | ------- | ------- | ------- | ------- | ------- | ------- | ------- | -------
> > | Seasonal Naive | CRPS | 0.070 | 0.512 | 0.085 | 0.515 | 0.205 | 0.257 | 0.294 | 0.151 | 0.068 | 0.262 | 1.000 | 1.000
> > |   | MASE | 0.881 | 1.203 | 0.906 | 1.778 | 1.390 | 1.137 | 1.669 | 1.236 | 0.782 | 0.986 | 1.000 | 1.000
> > | MoLE-DLinear | CRPS | 0.083 | 0.535 | 0.072 | 0.344 | 0.188 | 0.237 | 0.108 | 0.137 | 0.079 | 0.095 | 0.780 | 0.714
> > |   | MASE | 0.984 | 1.257 | 1.325 | 1.606 | 3.194 | 1.016 | 0.914 | 1.115 | 0.925 | 0.282 | 0.938 | 0.906
> > | TTM | CRPS | 0.075 | 0.534* | 0.059 | 0.417 | 0.122 | 0.210 | 0.150 | 0.192 | 0.055 | 0.102 | 0.758 | 0.697
> > |    | MASE | 0.802 | 1.255* | 0.898 | 1.934 | 1.547 | 0.901 | 1.195 | 1.477 | 0.506 | 0.308 | 0.831 | 0.798
> > | Timer | CRPS | 0.084 | 0.573 | 0.066 | 0.345 | 0.135 | 0.182 | 0.152 | 0.151 | 0.092 | 0.120 | 0.797 | 0.726
> > |    | MASE | 0.967 | 1.344 | 1.006 | 1.697 | 1.754 | 0.770 | 1.196 | 1.219 | 0.655 | 0.376 | 0.871 | 0.820
> > | Moment | CRPS | 0.354 | 1.332 | 0.151 | 0.401 | 0.277 | 0.612 | 0.157 | 0.154 | 0.105 | 0.313 | 1.502 | 1.205
> > |    | MASE | 3.167 | 3.139 | 2.244 | 2.243 | 4.100 | 2.617 | 1.277 | 1.245 | 1.053 | 0.913 | 1.691 | 1.457
> > | Time-MoE-Base | CRPS | 0.051* | 0.230* | 0.044 | 0.392 | 0.125 | 0.152 | 0.099 | 0.100 | 0.070 | 0.112 | 0.583 | 0.586
> > |    | MASE | 0.587* | 0.535* | 0.800 | 1.823 | 1.672 | 0.672 | 0.846 | 0.833 | 0.558 | 0.343 | 0.662 | 0.695
> > | Time-MoE-Large | CRPS | 0.051* | 0.294* | 0.045 | 0.386 | 0.131 | 0.172 | 0.090 | 0.097 | 0.058 | 0.111 | 0.589 | 0.576
> >  |   | MASE | 0.581* | 0.689* | 0.790 | 1.773 | 1.878 | 0.762 | 0.759 | 0.817 | 0.524 | 0.337 | 0.678 | 0.695
> > | Moirai-MoE-Small | CRPS | 0.046 | 0.429 | 0.036 | 0.288 | 0.093 | 0.108 | 0.071 | 0.090 | 0.056 | 0.081 | 0.497 | 0.450
> > |    | MASE | 0.719 | 1.222 | 0.737 | 1.750 | 1.248 | 0.563 | 0.746 | 0.927 | 0.476 | 0.298 | 0.670 | 0.620
> > Moirai-MoE-Base | CRPS | 0.041 | 0.382 | 0.034 | 0.296 | 0.091 | 0.100 | 0.071 | 0.088 | 0.057 | 0.079 | **0.478** | **0.439**
> > |    | MASE | 0.638 | 1.161 | 0.725 | 1.748 | 1.247 | 0.510 | 0.721 | 0.918 | 0.509 | 0.290 | **0.651** | **0.611**

---

> > > ### Author Response · Authors · 2024-11-22
> > > **Response to Reviewer xfCg (Part 3)**
> > >
> > > **[C3] The paper does not use established evaluation metrics MSE, MAE.**
> > >
> > > We report the MSE and MAE for zero-shot predictions below, and also in Appendix A, Table 8 of the revised paper (highlighted in red). We bold both the best and second best results here. Based on the results, Moirai-MoE achieves the best performance when considering the non-leak average values.
> > >
> > > | Method | Metric | Electricity | Solar | Power | ETT1 | ETT2 | Traffic | MDENSE | Walmart | Weather | BizITObs | Avg (all) | Avg (non-leak)
> > > | -------- | ------- | ------- | ------- | ------- | ------- | ------- | ------- | ------- | ------- | ------- | ------- | ------- | -------
> > > Seasonal Naive | MSE | 1299429.16 | 1293.24 | 1798196.83 | 57976.63 | 122878.95 | 203.32 | 39929.67 | 32876026.66 | 2197.23 | 174.31 | 1.000 | 1.000
> > > | | MAE | 166.20 | 15.77 | 492.60 | 154.98 | 211.56 | 8.72 | 118.38 | 2637.43 | 10.96 | 9.69 | 1.000 | 1.000
> > > iTransformer | MSE | 1264494.38 | 1183.57 | 968959.56 | 55320.57 | 178757.02 | 41.77 | 9905.39 | 10922819.00 | 1885.01 | 20.55 | 0.508 | 0.435
> > > | | MAE | 165.89 | 17.61 | 399.09 | 170.83 | 279.21 | 4.85 | 51.06 | 1560.68 | 10.65 | 2.66 | 0.741 | 0.678
> > > MoLE-DLinear | MSE | 1901617.97 | 1098.56 | 1071490.46 | 39026.37 | 195287.19 | 153.71 | 13016.78 | 26832049.08 | 1649.90 | 21.57 | 0.656 | 0.575
> > > | | MAE | 197.06 | 16.47 | 420.67 | 130.79 | 328.28 | 8.48 | 62.43 | 2395.50 | 12.81 | 2.75 | 0.857 | 0.803
> > > TimesFM | MSE | 1378828.95* | 1061.70 | 384815.80 | 42789.02 | 169714.41 | 106.01 | 10194.73 | 9494507.86 | 1317.09 | 23.23 | 0.475 | 0.401
> > > | | MAE | 137.57* | 18.07 | 277.94 | 138.42 | 245.61 | 5.75 | 49.78 | 1484.68 | 7.94 | 2.89 | 0.672 | **0.612**
> > > TTM | MSE | 2432897.66 | 884.33* | 647289.67 | 56256.46 | 116203.30 | 114.79 | 18425.62 | 39297380.00 | 1122.55 | 23.41 | 0.625 | 0.538
> > > | | MAE | 179.56 | 16.46* | 341.96 | 158.85 | 213.61 | 7.53 | 86.44 | 3360.79 | 8.88 | 2.97 | 0.833 | 0.784
> > > Timer | MSE | 2205084.30 | 962.26 | 687600.25 | 39235.36 | 129063.67 | 75.23 | 19875.60 | 29410540.00 | 1873.68 | 27.21 | 0.613 | 0.527
> > > | | MAE | 200.62 | 17.57 | 370.53 | 131.31 | 235.27 | 6.42 | 87.72 | 2646.92 | 13.65 | 3.50 | 0.865 | 0.804
> > > Moment | MSE | 44303358.90 | 2876.47 | 3272382.39 | 46075.47 | 411967.28 | 601.62 | 19506.54 | 29046437.85 | 1804.48 | 129.26 | 1.760 | 1.180
> > > | | MAE | 843.45 | 41.02 | 873.48 | 152.56 | 484.86 | 21.87 | 90.51 | 2690.84 | 16.89 | 9.11 | 1.650 | 1.355
> > > Chronos-Small | MSE | 1251170.49* | 1405.10* | 418195.72 | 60157.02 | 112472.02 | 100.62 | 15377.29 | 14697271.28 | 3945.04 | 23.89 | 0.587 | 0.511
> > > | | MAE | 126.25* | 15.79* | 275.11 | 161.23 | 207.11 | 5.28 | 59.26 | 1693.33 | 16.90 | 2.94 | 0.724 | 0.691
> > > Chronos-Base | MSE | 1147348.35* | 1062.73* | 400709.37 | 66320.26 | 107178.21 | 80.48 | 12770.66 | 15813384.14 | 1720.53 | 22.78 | 0.501 | 0.439
> > > | | MAE | 121.69* | 13.18* | 285.79 | 169.60 | 194.70 | 4.69 | 51.58 | 1706.11 | 10.28 | 2.82 | 0.656 | 0.628
> > > Chronos-Large | MSE | 1073679.39* | 1017.98* | 362386.33 | 73974.48 | 106362.90 | 98.20 | 13625.07 | 12339319.84 | 1874.83 | 23.61 | 0.503 | 0.447
> > > | | MAE | 121.06* | 12.86* | 277.64 | 177.68 | 191.07 | 5.07 | 53.61 | 1560.11 | 11.30 | 2.89 | 0.664 | 0.639
> > > Moirai-Small | MSE | 4015423.50 | 1429.82 | 757613.06 | 39481.46 | 118636.33 | 146.24 | 11041.41 | 19886286.00 | 1932.16 | 22.48 | 0.647 | 0.498
> > > | | MAE | 219.02 | 19.19 | 358.01 | 133.82 | 209.68 | 8.71 | 58.25 | 2112.07 | 10.23 | 2.90 | 0.802 | 0.715
> > > Moirai-Base | MSE | 1734656.25 | 1105.95 | 477193.47 | 51793.64 | 113074.23 | 44.60 | 17724.71 | 18981036.00 | 1196.21 | 22.44 | 0.500 | 0.414
> > > | | MAE | 164.94 | 16.97 | 293.74 | 149.15 | 202.89 | 4.72 | 79.41 | 2046.22 | 7.73 | 2.81 | 0.713 | 0.650
> > > Moirai-Large | MSE | 1229872.00 | 997.13 | 340307.44 | 44752.48 | 106513.38 | 101.17 | 14874.89 | 21274060.00 | 1914.39 | 21.79 | 0.511 | 0.449
> > > | | MAE | 150.66 | 16.25 | 262.70 | 142.21 | 204.72 | 5.93 | 69.73 | 2110.73 | 10.10 | 2.77 | 0.720 | 0.669
> > > Time-MoE-Base | MSE | 1158323.38* | 176.27* | 315704.91 | 50267.22 | 114374.42 | 89.87 | 11303.31 | 13934856.92 | 1371.87 | 28.51 | **0.395** | 0.408
> > > | | MAE | 120.52* | 7.07* | 254.28 | 149.21 | 218.55 | 5.70 | 57.43 | 1742.96 | 11.35 | 3.26 | **0.644** | 0.663
> > > Time-MoE-Large | MSE | 1203643.75* | 194.84* | 350989.67 | 47389.70 | 121112.59 | 99.13 | 9585.73 | 12876789.32 | 1264.26 | 27.34 | **0.394** | 0.400
> > > | | MAE | 120.53* | 9.06* | 262.48 | 147.11 | 229.67 | 6.45 | 52.10 | 1687.08 | 9.32 | 3.24 | 0.650 | 0.652
> > > Moirai-MoE-Small | MSE | 930140.63 | 1113.50 | 360995.59 | 45412.81 | 114609.09 | 53.05 | 9426.45 | 18025986.00 | 1944.27 | 23.45 | 0.453 | **0.395**
> > > | | MAE | 138.03 | 16.05 | 260.82 | 141.08 | 194.63 | 4.78 | 50.09 | 1955.77 | 10.08 | 2.89 | 0.668 | 0.617
> > > Moirai-MoE-Base | MSE | 907276.31 | 1047.63 | 311227.06 | 48487.21 | 107284.42 | 45.83 | 9740.51 | 17094764.00 | 1954.24 | 22.54 | 0.434 | **0.378**
> > > | | MAE | 122.27 | 15.24 | 251.10 | 145.50 | 191.47 | 4.33 | 49.73 | 1919.31 | 10.31 | 2.80 | **0.646** | **0.605**

---

> > > > ### Author Response · Authors · 2024-11-22
> > > > **Response to Reviewer xfCg (Part 4)**
> > > >
> > > > **[C4] Could the authors provide detailed experimental setups, including the input and prediction lengths for each model across the datasets, as well as a comparison table with previous mainstream time series prediction works? Specifically, the prediction lengths should be (96, 192, 336, 720) to facilitate intuitive comparisons for reviewers.**
> > > >
> > > > (1) Thank you for raising this concern. The prediction lengths for the 39 datasets are already detailed in Appendix A (Table 5 and Table 7). We apologize for not providing the input lengths for each model. To address this, we present a table below detailing the input lengths used for each method in this study, and in their original paper. For full-shot deep learning models, we believe our searching range generally covers the lengths set in their original paper. For foundation models, the choice of input lengths depends on their pretraining strategies. For instance, in the case of TimesFM and Chronos, the input lengths are consistently set to 512 during pretraining. In contrast, for Moirai and Moirai-MoE, the pretraining algorithm involves randomly sampling a context length in the range [2, 8192]. Thus, searching for the input length on validation set during inference is needed. We have added this part in Table 9 of the revised paper (highlighted in red).
> > > >
> > > > | Method | In-Dist. Evaluation (29 datasets) | Zero-Shot Evaluation (10 datasets) | Original Paper
> > > > | -------- | ------- | ------- | -------
> > > > TiDE | -- | Searching within prediction lengths * [2,20] | 720
> > > > PatchTST | -- | Searching within prediction lengths * [2,20] | 336
> > > > iTransformer | -- | Searching within prediction lengths * [2,20] | 96
> > > > TTM | -- | 512 | 512
> > > > Timer | -- | 672 | 672
> > > > Moment | -- | 512 | 512
> > > > Time-MoE | -- | 4096 | \{512, 1024, 2048, 3072\}
> > > > TimesFM | 512 | 512 | 512
> > > > Chronos | 512 | 512 | 512
> > > > Moirai | 1000 | Searching within range \{1000, 2000, 3000, 4000, 5000\} | Searching within range \{1000, 2000, 3000, 4000, 5000\}
> > > > Moirai-MoE | 1000 | Searching within range \{1000, 2000, 3000, 4000, 5000\} | Searching within range \{1000, 2000, 3000, 4000, 5000\}
> > > >
> > > > (2) Regarding the comparison table with previous mainstream works, we believe we have included the most prominent and competitive deep learning methods: TiDE, PatchTST, and iTransformer. Additionally, we incorporated foundation models such as TimesFM, Chronos, and Moirai. During the rebuttal stage, we further expanded our comparison by including the deep learning model MoLE-DLinear and foundation models Timer, Moment, TTM, Time-MoE-Base, and Time-MoE-Large.
> > > >
> > > > (3) Regarding the evaluation data and prediction lengths, we argue that the benchmarks set by previous studies—specifically the datasets ETTh1 (H), ETTh2 (H), ETTm1 (15T), ETTm2 (15T), Electricity (H), Weather (10T), Traffic (H), and Solar (10T)—exhibit a strong bias toward the electricity data and the frequencies of hourly and minutely. Additionally, ETTh1 and ETTm1, as well as ETTh2 and ETTm2, are essentially derived from the same underlying data. Furthermore, some of these datasets were already part of our pretraining corpus, raising potential data leakage concerns.
> > > >
> > > > To address these issues and establish a more diverse zero-shot evaluation scenario without data leakage, we adopt six zero-shot datasets from the Moirai paper and incorporate four additional datasets (details in Appendix A, Table 7). These added datasets enhance the diversity of our evaluation, making our evaluation cover five domains and include frequencies ranging from minute-level to weekly. However, due to the inherent nature of some datasets, such as limited test set lengths, it is not feasible to apply the same prediction length uniformly across all these data. Consequently, the prediction lengths vary across datasets in this study.
> > > >
> > > > **To be continued ...**

---

> > > > > ### Author Response · Authors · 2024-11-22
> > > > > **Response to Reviewer xfCg (Part 5)**
> > > > >
> > > > > As detailed in Tables 5 and 7, the prediction lengths of our evaluation data already span a wide range, from 8 to 336. To address your concern regarding the prediction length, we have conducted additional experiments using a prediction length of 720 on the Informer benchmarks: ETTh1, ETTm1, ETTh2, ETTm2, and Weather datasets. The results are presented below, and we bold both the best and second best results. We have the following observations. First, the full-shot deep learning methods deliver the best overall performance. Second, Moirai-MoE cannot beat Moirai at the prediction length of 720, which we attribute to the error accumulation inherent in decoder-only architectures. Addressing this limitation is left for future work. Third, Moirai-MoE outperforms other decoder-only-based foundation models such as TimesFM, Timer, Time-MoE, and Chronos, in terms of CRPS and MASE.
> > > > >
> > > > > | Method | ETTm1 |  | ETTh1 | | ETTm2 | | ETTh2 |  | Weather | | Avg | |
> > > > > | -------- | ------- | ------- | ------- | ------- | -------  | ------- | -------  | ------- | -------  | -------  | -------  | -------
> > > > > | | CRPS | MASE | CRPS | MASE | CRPS | MASE | CRPS | MASE | CRPS | MASE | CRPS | MASE
> > > > > Seasonal Naive | 0.396 | 1.191 | 0.616 | 1.479 | 0.165 | 1.013 | 0.287 | 1.128 | 0.304 | 0.761 | 1.000 | 1.000
> > > > > TiDE | 0.288 | 1.217 | 0.313 | 1.520 | 0.127 | 1.390 | 0.128 | 1.312 | 0.085 | 0.822 | 0.512 | 1.126
> > > > > PatchTST | 0.247 | 1.098 | 0.297 | 1.468 | 0.098 | 0.961 | 0.130 | 1.433 | 0.066 | 1.074 | **0.446** | 1.093
> > > > > iTransformer | 0.245 | 1.110 | 0.297 | 1.432 | 0.101 | 1.006 | 0.125 | 1.332 | 0.064 | 0.690 | **0.442** | **0.992**
> > > > > TimesFM | 0.358 | 1.316 | 0.317 | 1.408 | 0.119 | 1.034 | 0.125 | 1.090 | 0.069 | 1.359 | 0.506 | 1.131
> > > > > TTM | 0.474 | 1.488 | 0.357 | 1.394 | 0.139 | 1.083 | 0.153 | 1.219 | 0.069 | 0.674 | 0.589 | 1.038
> > > > > Timer | 0.562 | 1.727 | 0.440 | 1.608 | 0.146 | 1.143 | 0.152 | 1.198 | 0.103 | 0.738 | 0.694 | 1.129
> > > > > Moment | 0.555 | 1.729 | 0.572 | 1.977 | 0.149 | 1.159 | 0.173 | 1.380 | 0.108 | 0.766 | 0.759 | 1.223
> > > > > Chronos-Small | 0.450 | 1.496 | 0.360 | 1.424 | 0.129 | 1.111 | 0.122 | 1.041 | 0.096 | 0.956 | 0.587 | 1.090
> > > > > Chronos-Base | 0.400 | 1.349 | 0.350 | 1.430 | 0.134 | 1.137 | 0.136 | 1.121 | 0.080 | 0.802 | 0.565 | 1.052
> > > > > Chronos-Large | 0.367 | 1.282 | 0.350 | 1.463 | 0.140 | 1.168 | 0.139 | 1.136 | 0.077 | 0.756 | 0.559 | 1.042
> > > > > Moirai-Small | 0.259 | 1.142 | 0.313 | 1.545 | 0.105 | 1.066 | 0.115 | 1.160 | 0.067 | 0.680 | 0.452 | **0.993**
> > > > > Moirai-Base | 0.273 | 1.123 | 0.287 | 1.382 | 0.137 | 1.304 | 0.110 | 1.121 | 0.071 | 0.762 | 0.473 | 1.024
> > > > > Moirai-Large | 0.358 | 1.403 | 0.296 | 1.453 | 0.115 | 1.139 | 0.125 | 1.281 | 0.077 | 0.792 | 0.508 | 1.090
> > > > > Time-MoE-Base | 0.613 | 2.005 | 0.581 | 1.880 | 0.196 | 1.509 | 0.202 | 1.869 | 0.196 | 1.269 | 0.953 | 1.545
> > > > > Time-MoE-Large | 0.487 | 1.643 | 0.419 | 1.545 | 0.170 | 1.321 | 0.162 | 1.398 | 0.113 | 0.954 | 0.711 | 1.239
> > > > > Moirai-MoE-Small | 0.404 | 1.422 | 0.324 | 1.462 | 0.105 | 1.032 | 0.117 | 1.145 | 0.076 | 0.660 | 0.510 | 1.011
> > > > > Moirai-MoE-Base | 0.397 | 1.475 | 0.314 | 1.399 | 0.103 | 1.031 | 0.108 | 1.073 | 0.075 | 0.689 | 0.495 | 1.005
> > > > >
> > > > > **To be continued ...**

---

> ### Author Response · Authors · 2024-11-22
> **Response to Reviewer xfCg (Part 6)**
>
> | Method | ETTm1 |  | ETTh1 | | ETTm2 | | ETTh2 |  | Weather | | Avg | |
> | -------- | ------- | ------- | ------- | ------- | -------  | ------- | -------  | ------- | -------  | -------  | -------  | -------
> | | MSE | MAE | MSE | MAE | MSE | MAE | MSE | MAE | MSE | MAE | MSE | MAE
> Seasonal Naive | 13.023 | 1.856 | 199.495 | 7.749 | 13.675 | 2.362 | 258.848 | 10.012 | 2044.538 | 11.442 | 1.000 | 1.000
> TiDE | 10.255 | 1.878 | 178.306 | 8.230 | 16.388 | 2.917 | 303.572 | 11.229 | 1611.362 | 13.609 | **0.951** | 1.121
> PatchTST | 8.698 | 1.670 | 168.882 | 7.972 | 12.485 | 2.250 | 319.476 | 12.044 | 1869.367 | 13.002 | **0.898** | **1.038**
> iTransformer | 8.191 | 1.651 | 180.740 | 7.905 | 13.418 | 2.398 | 316.962 | 11.444 | 2531.117 | 12.462 | 0.967 | **1.028**
> TimesFM | 12.573 | 2.111 | 179.730 | 7.400 | 14.416 | 2.440 | 250.114 | 10.043 | 1923.572 | 12.344 | 0.964 | 1.040
> TTM | 16.588 | 2.484 | 149.526 | 7.463 | 15.252 | 2.601 | 288.521 | 11.162 | 1562.930 | 11.357 | 0.981 | 1.095
> Timer | 26.456 | 2.947 | 252.667 | 9.197 | 17.283 | 2.730 | 318.642 | 11.128 | 2642.381 | 16.858 | 1.389 | 1.290
> Moment | 24.626 | 2.911 | 392.801 | 11.937 | 17.587 | 2.794 | 383.546 | 12.609 | 2408.585 | 17.682 | 1.529 | 1.409
> Chronos-Small | 19.216 | 2.463 | 193.029 | 7.772 | 17.203 | 2.572 | 226.874 | 9.395 | 3306.816 | 17.162 | 1.206 | 1.153
> Chronos-Base | 15.446 | 2.199 | 173.446 | 7.624 | 21.584 | 2.650 | 274.234 | 10.419 | 2477.497 | 14.173 | 1.159 | 1.110
> Chronos-Large | 13.662 | 2.014 | 165.268 | 7.614 | 23.625 | 2.752 | 288.807 | 10.601 | 2391.803 | 13.652 | 1.144 | 1.094
> Moirai-Small | 9.871 | 1.742 | 172.307 | 8.375 | 15.227 | 2.546 | 277.093 | 10.687 | 2650.900 | 12.753 | 1.002 | 1.054
> Moirai-Base | 11.349 | 1.801 | 156.355 | 7.475 | 30.463 | 3.053 | 257.796 | 10.097 | 2662.350 | 13.135 | 1.146 | 1.070
> Moirai-Large | 22.010 | 2.314 | 171.837 | 7.727 | 19.239 | 2.739 | 351.780 | 11.825 | 2562.314 | 14.319 | 1.284 | 1.163
> Time-MoE-Base | 29.783 | 3.218 | 414.974 | 12.141 | 29.791 | 3.670 | 452.066 | 14.755 | 6514.563 | 32.020 | 2.250 | 1.771
> Time-MoE-Large | 19.818 | 2.555 | 214.722 | 8.758 | 24.203 | 3.189 | 324.834 | 11.855 | 2798.035 | 18.487 | 1.379 | 1.321
> Moirai-MoE-Small | 16.175 | 2.260 | 188.150 | 7.726 | 12.938 | 2.363 | 290.732 | 10.568 | 2742.873 | 13.164 | 1.108 | 1.081
> Moirai-MoE-Base | 23.064 | 2.417 | 193.842 | 7.516 | 13.556 | 2.389 | 256.354 | 9.634 | 2746.904 | 13.243 | 1.178 | 1.073
>
> **[C5] In Appendix, the ablation study on patch size reveals that it significantly affects performance. This raises the question of whether multi-patch projection is indeed an effective strategy. Could the authors explain why patch size still significantly affects the model, and whether multi-patch projection remains necessary under this phenomenon?**
>
> Thank you for your thoughtful feedback. It seems there may be some misunderstanding. Moirai uses multi-patch projection for each frequency, whereas Moirai-MoE does not employ multi-patch projection. In Moirai-MoE, we employ single-patch projection and the patch size is a single hyperparameter that can be tuned. This is a common approach in patch-based models, such as TimesFM.
>
> Regarding the choice of patch size in relation to performance, the patch size determines the time period encompassed within each token. If the patch size is too large, the linear projection layer may lack the capacity to capture the underlying patterns. Conversely, if the patch size is too small, the time series token may not contain sufficient semantic information, as highlighted in DLinear.
>
> **[C6] Could the authors provide insight into whether the token clusters generated by different time series foundational models affect the performance of MOIRAI-MOE?**
>
> Thank you for this insightful comment. We have conducted additional experiments, the details of which are outlined below. Specifically, we utilized the Chronos-Small checkpoint, as it matches the number of Transformer layers in Moirai-MoE-Small. Similar to Moirai-MoE, we performed inference on Chronos using our pretraining dataset, LOTSA, to extract the hidden states after the attention modules and generate cluster centroids at each layer. Subsequently, we pretrained a Moirai-MoE-Chronos variant using the token clusters derived from Chronos. The results on Monash, presented below, show that the performance of this variant is comparable to Moirai-MoE, demonstrating that our proposed gating function is capable of leveraging knowledge from other foundation models.
>
> | Variant | Pretrained Model | Inference Data for Cluster Generation| Layers | d_ff | Performance
> | -------- | ------- | ------- | -------  | ------- | -------
> Moirai-MoE-Chronos | Chronos-Small | LOTSA | 6 | 512 | 0.67
> Moirai-MoE | Moirai-Small | LOTSA | 6 | 512 | 0.65
>
> We are very grateful for your insightful comments. Your expert knowledge is helping us to strengthen the manuscript significantly. Please let us know if you have any other comments/questions.

---

> > ### Author Response · Authors · 2024-11-25
> > **Kindly Request for Reviewer's Feedback**
> >
> > Dear reviewer xfCg,
> >
> > Thank you for taking the time and effort in providing a valuable review of our work. As the discussion period is coming to a close, we hope that you have had the chance to review our rebuttal. We believe that our rebuttal has provided additional clarity of our work. If our response has addressed your concerns, we hope that you could kindly leave an update to reflect this, and are more than willing to engage in further discussion if needed.

---

> ### Comment · Reviewer_xfCg · 2024-11-25
> **Response to authors**
>
> Thank you very much for your response, which has addressed most of my concerns. However, there are still a few questions that remain unresolved:
>
> - I would like to know the performance of MOIRAI-MOE in both few-shot and full-shot settings, and how it compares using standard benchmarking methods, such as the settings in PatchTST.
>
> - Additionally, I am curious whether excessively long input lengths could lead to unfair comparisons. For instance, if the input length reaches as high as 5000 as shown in your table, specific models trained on such extensive data might also perform quite well, which could diminish the advantage of foundational time series models.
>
> - Lastly, regarding the multi-patch strategy, I may not have expressed myself clearly. What I meant is that if the patch size has a significant impact on the performance of MOIRAI-MOE, does that imply that MOIRAI-MOE has not fully addressed the lack of generalizability associated with a single patch size? In that case, the original multi-patch strategy in MOIRAI should still hold positive significance.

---

> ### Author Response · Authors · 2024-11-28
> **Response to Reviewer xfCg (Part 7)**
>
> Dear reviewer, thank you much for your reply. Please find our responses below.
>
> **[C7] I would like to know the performance of MOIRAI-MOE in both few-shot and full-shot settings, and how it compares using standard benchmarking methods, such as the settings in PatchTST.**
>
> Thanks for the comment. Knowing the performance of foundation models in fine-tuning downstream datasets under few-shot or full-shot settings is valuable; however, this requirement falls outside the scope of our study. The main goal of this work lies in developing time series foundation models that can generalize well in zero-shot forecasting scenarios [1,2,3].
>
> And we have conducted extensive experiments to support our claim. We included 10 zero-shot datasets spanning diverse domains and sampling frequencies. We benchmarked these datasets against 11 baseline models, and, in response to the reviewer's request, added results for 6 additional baselines, bringing the total to 17. Most importantly, Moirai-MoE outperformed all 17 baselines, showcasing its superiority in zero-shot forecasting. We leave the investigation of fine-tuning foundation models into future work.
>
> [1] Unified Training of Universal Time Series Forecasting Transformers.
>
> [2] Chronos: Learning the Language of Time Series.
>
> [3] A decoder-only foundation model for time-series forecasting.
>
> **[C8] I am curious whether excessively long input lengths could lead to unfair comparisons. For instance, if the input length reaches as high as 5000 as shown in your table, specific models trained on such extensive data might also perform quite well, which could diminish the advantage of foundational time series models.**
>
> Thank you for raising this point. If we understand correctly, you are suggesting that training a specialized model on a dataset with a long input length, such as 5000, could achieve strong performance or even surpass foundational models. To investigate this and also to address your concern about the potential unfair comparisons arising from different context lengths, we have conducted additional experiments: training the state-of-the-art specialized models PatchTST and iTransformers using the same context length searching strategy as in Moirai and Moirai-MoE.
>
> The below table presents the results using 4 metrics. Please note that, for clarity, we use Range-1 to denote the searching range within prediction lengths * [2, 20], which is approximately around the values of [50, 500]. This is the range we used in our paper for specialized models. In addition, we use Range-2 to denote Moirai's strategy: searching within {1000, 2000, 3000, 4000, 5000}.
>
> | Method | Context Length | Metric | Electricity | Solar | Power | ETT1 | ETT2 | Traffic | MDENSE | Walmart | Weather | BizITObs | Avg (all)
> | -------- | ------- | ------- | ------- | ------- | ------- | ------- | ------- | ------- | ------- | ------- | ------- | ------- | -------
> | Seasonal Naive | -- | CRPS | 0.070 | 0.512 | 0.085 | 0.515 | 0.205 | 0.257 | 0.294 | 0.151 | 0.068 | 0.262 | 1.000 | 1.000
> |   | -- | MASE | 0.881 | 1.203 | 0.906 | 1.778 | 1.390 | 1.137 | 1.669 | 1.236 | 0.782 | 0.986 | 1.000 | 1.000
> | PatchTST | Range-1 | CRPS | 0.052 | 0.518 | 0.054 | 0.304 | 0.131 | 0.112 | 0.070 | 0.082 | 0.059 | 0.074 | 0.549
> |  |  | MASE | 0.753 | 1.607 | 1.234 | 1.680 | 2.168 | 0.653 | 0.732 | 0.867 | 0.844 | 0.266 | 0.808
> | PatchTST | Range-2 | CRPS | 0.050 | 0.389 | 0.057 | 0.551 | 0.154 | 0.110 | 0.149 | 0.092 | 0.080 | 0.066 | 0.640
> |  |  | MASE | 0.736 | 1.194 | 1.125 | 3.528 | 2.647 | 0.594 | 1.552 | 0.888 | 1.659 | 0.244 | 0.967
> | iTransformer | Range-1 | CRPS | 0.057 | 0.443 | 0.056 | 0.344 | 0.129 | 0.105 | 0.072 | 0.070 | 0.053 | 0.077 | 0.540
> |  |  | MASE | 0.875 | 1.342 | 1.076 | 2.393 | 1.841 | 0.581 | 0.727 | 0.761 | 0.623 | 0.271 | 0.767
> | iTransformer | Range-2 | CRPS | 0.058 | 0.607 | 0.047 | 0.345 | 0.190 | 0.109 | 0.117 | 0.111 | 0.053 | 0.148 | 0.671
> |  |  | MASE | 0.828 | 1.828 | 1.018 | 2.302 | 3.687 | 0.568 | 1.275 | 1.160 | 0.766 | 0.581 | 1.013
> | Moirai-MoE-Base | Range-2 | CRPS | 0.041 | 0.382 | 0.034 | 0.296 | 0.091 | 0.100 | 0.071 | 0.088 | 0.057 | 0.079 | **0.478**
> |  |  | MASE | 0.638 | 1.161 | 0.725 | 1.748 | 1.247 | 0.510 | 0.721 | 0.918 | 0.509 | 0.290 | **0.651**
>
> **To be continued ...**

---

> > ### Author Response · Authors · 2024-11-28
> > **Response to Reviewer xfCg (Part 8)**
> >
> > | Method | Context Length | Metric | Electricity | Solar | Power | ETT1 | ETT2 | Traffic | MDENSE | Walmart | Weather | BizITObs | Avg (all)
> > | -------- | ------- | ------- | ------- | ------- | ------- | ------- | ------- | ------- | ------- | ------- | ------- | ------- | -------
> > | Seasonal Naive | -- | MSE | 1299429.16 | 1293.24 | 1798196.83 | 57976.63 | 122878.95 | 203.32 | 39929.67 | 32876026.66 | 2197.23 | 174.31 | 1.000
> > | | -- | MAE | 166.20 | 15.77 | 492.60 | 154.98 | 211.56 | 8.72 | 118.38 | 2637.43 | 10.96 | 9.69 | 1.000
> > | PatchTST | Range-1 | MSE | 1534813.00 | 1125.03 | 1605878.50 | 45755.70 | 167280.13 | 38.69 | 9240.59 | 13749435.00 | 1361.20 | 18.97 | 0.511
> > | | | MAE | 171.95 | 17.03 | 478.86 | 141.03 | 291.80 | 4.51 | 49.45 | 1847.65 | 9.77 | 2.60 | 0.740
> > | PatchTST | Range-2 | MSE | 1433466.75 | 973.65 | 957680.50 | 249211.48 | 284311.13 | 45.14 | 38315.62 | 15738720.00 | 2020.57 | 15.18 | 0.717
> > | | | MAE | 150.33 | 15.69 | 414.32 | 259.93 | 346.99 | 4.96 | 111.85 | 1859.13 | 15.96 | 2.36 | 0.880
> > | iTransformer | Range-1 | MSE | 1264494.38 | 1183.57 | 968959.56 | 55320.57 | 178757.02 | 41.77 | 9905.39 | 10922819.00 | 1885.01 | 20.55 | 0.508
> > | | | MAE | 165.89 | 17.61 | 399.09 | 170.83 | 279.21 | 4.85 | 51.06 | 1560.68 | 10.65 | 2.66 | 0.741
> > | iTransformer | Range-2 | MSE | 1856541.38 | 2587.88 | 572127.56 | 54057.63 | 297574.38 | 39.58 | 19981.01 | 21888656.00 | 1375.14 | 50.39 | 0.689
> > | | | MAE | 174.00 | 23.96 | 350.17 | 167.26 | 408.40 | 4.74 | 89.69 | 2504.30 | 10.11 | 5.53 | 0.930
> > | Moirai-MoE-Base | Range-2 | MSE | 907276.31 | 1047.63 | 311227.06 | 48487.21 | 107284.42 | 45.83 | 9740.51 | 17094764.00 | 1954.24 | 22.54 | **0.434**
> > | | | MAE | 122.27 | 15.24 | 251.10 | 145.50 | 191.47 | 4.33 | 49.73 | 1919.31 | 10.31 | 2.80 | **0.646**
> >
> > According to the results, we observe that from Range-1 to Range-2, the input length searching range increases substantially. However, the average performance of both PatchTST and iTransformer declines significantly across all four metrics. For instance, the average MASE of PatchTST increases from 0.808 to 0.967, and the average MASE of iTransformer rises from 0.767 to 1.013. These results suggest that neither PatchTST nor iTransformer benefits from very long input lengths; in fact, such lengths negatively impact their performance. And the performance gap between these models and the foundation models becomes even more pronounced.
> >
> > Regarding your argument about the good performance of specialized models trained with long inputs, e.g., 5000, may undermine the advantages of time series foundation models. We respectfully disagree with this. First, our experiments indicate that using excessively long inputs often results in degraded performance, counter to this claim. Second, while it is possible that certain specialized models not evaluated here could perform well with long inputs on specific datasets, the core motivation for time series foundation models lies in their ability to provide a unified framework for diverse downstream forecasting tasks. The success of specialized models on isolated datasets does not detract from the broader advantages and applicability of foundation models.
> >
> > **[C9] If the patch size has a significant impact on the performance of MOIRAI-MOE, does that imply that MOIRAI-MOE has not fully addressed the lack of generalizability associated with a single patch size? In that case, the original multi-patch strategy in MOIRAI should still hold positive significance.**
> >
> > Thank you for the clarification. To address your concern, we have conducted additional experiments, and the results are presented in the table below. The notation used is as follows: MoE-Small-Multi refers to Moirai-MoE-Small with a multi-patch strategy. MoE-Small-S4 represents Moirai-MoE-Small with a single patch size of 4. The same naming applies to other configurations.
> >
> > The results show that S4 and S64 do not perform as well as S16, however, they significantly outperform Moirai-Small and achieve comparable results to MoE-Small-Muiti. These results indicate that the multi-patch strategy does not hold positive significance, since it is comparable to the worst cases of single patch size.
> >
> > | Method | Moirai-Small | MoE-Small-Multi | MoE-Small-S4 | MoE-Small-S8 | MoE-Small-S16 | MoE-Small-S32 | MoE-Small-S64
> > | -------- | ------- | ------- | ------- | ------- | ------- | ------- | -------
> > | Monash | 0.78 | 0.72 | 0.72 | 0.67 | 0.65 | 0.70 | 0.72
> >
> > Regarding the choice of patch size in relation to performance, the patch size determines the time period encompassed within each token. If the patch size is too large (i.e., 64), the linear projection layer may lack the capacity to capture the underlying patterns. Conversely, if the patch size is too small (i.e., 4), the time series token may not contain sufficient semantic information, as highlighted in DLinear.
> >
> > Thank you again for your constructive feedback. We hope our clarifications address your concerns.

---

> > > ### Comment · Reviewer_xfCg · 2024-12-01
> > >
> > > I have improved scores. Good luck.

---

> > > > ### Author Response · Authors · 2024-12-01
> > > > **Thank you for your support**
> > > >
> > > > Dear reviewer,
> > > >
> > > > Thank you so much for your reply. We are thrilled to hear that our response has addressed your concerns. We would like to express our sincerest gratitude once again for taking the time to review our paper and providing us with invaluable suggestions.

---

### Official Review · Reviewer_R7WZ · 2024-11-02

**Soundness:** 2
**Presentation:** 3
**Contribution:** 2
**Rating:** 5
**Confidence:** 3

**Summary:**

This paper introduces MOIRAI-MOE, a time series foundation model that overcomes limitations of frequency-based specialization by using a sparse Mixture of Experts (MoE) architecture within Transformers, allowing token-level, data-driven specialization. A novel gating function based on cluster centroids improves expert allocation accuracy, reducing reliance on human heuristics. MOIRAI-MOE outperforms existing models across the 39 testing datasets, including its predecessor MOIRAI. It generalizes effectively across varied time series patterns, with shallow layers capturing short-term variability and deeper layers learning more stable, abstract patterns.

**Strengths:**

1. MOIRAI-MOE’s mixture-of-experts design for time series reduces reliance on frequency-based specialization by using sparse experts for flexible, data-driven adaptation.

2. This work proposed an automatic, token-level specialization that allows the model to handle diverse time series patterns, enhancing flexibility and robustness.

3. This work introduced a gating function based on pre-trained cluster centroids that can potentially improve expert selection accuracy.

**Weaknesses:**

Writing:

1. The token clustering mechanism used as a gating function, intended as a key differentiator from existing work, is only briefly described in about ten lines. Please consider spending more text on the advantages of the proposed gating function.

Experimentation:

2. Section 4.3 does not sufficiently demonstrate the irreplaceability of token clusters in the gating function.

3. In Figure 6, from layer 4 onward, different tokens tend to route to the same experts (three experts in layer 4 and two in layer 6), suggesting that the model may have more experts than necessary for practical use.

4. If MOIRAI-MOE is indeed a promising architecture, it should demonstrate lower MSE as its parameter count scales up. However, Table 2 lacks sufficient evidence for this, as it only includes the MoE-Base model with 86M/935M parameters.

Reproducibility:

5. The team has not open-sourced the code for evaluation, impacting the reproducibility of this work.

**Questions:**

Please refer to the weakness part.

---

> ### Author Response · Authors · 2024-11-22
> **Response to Reviewer R7WZ**
>
> Dear reviewer, we would like to express our sincere gratitude for the time you have invested in reviewing our submission, and thank you very much for appreciating the model designs of Moirai-MoE. Please find our responses below.
>
> **[C1] Please consider spending more text on the advantages of the proposed gating function.**
>
> Thanks for the comment. We have revised Section 3.2.1 of the paper to address this point, with the changes highlighted in red.
>
> **[C2] Section 4.3 does not sufficiently demonstrate the irreplaceability of token clusters in the gating function.**
>
> Thank you for pointing this out. We would like to clarify that we do not claim or emphasize the irreplaceability of token clusters in this study. The primary focus of our work is to address the challenges of achieving effective unified training for time series, and we propose leveraging MoE to enable automatic token-level model specialization.
>
> A natural question when employing MoE is how to design the gating function. We have explored various gating mechanisms and found that token clustering performs best for time series prediction. However, we do not assert that it is irreplaceable. To further illustrate this point, the table below presents a performance comparison on the Monash benchmark. As shown, the linear-projection-based MoE can already offer competitive performance and the token-clusters-based MoE further improves it. Therefore, the superiority of Moirai-MoE stems from the use of MoE, and the token clusters method can help achieve further enhancements.
>
> | Method | TimesFM | Chronos-Small | Chronos-Base | Chronos-Large | Moirai-Small | Moirai-Base | Moirai-Large | Moirai-MoE-Small-Linear | Moirai-MoE-Small-Clusters
> | -------- | ------- | ------- | ------- | ------- | ------- | ------- | ------- | ------- | -------
> | Monash | 0.77 | 0.67 | 0.71 | 0.72 | 0.78 | 0.71 | 0.70 | 0.67  | 0.65
>
> **[C3] In Figure 6, from layer 4 onward, different tokens tend to route to the same experts, suggesting that the model may have more experts than necessary for practical use.**
>
> Thank you for this insightful comment. We agree with the reviewer that some experts at deep layers in Moirai-MoE are rarely selected during inference on the datasets from Monash. Pruning these underutilized experts to improve inference efficiency is an interesting direction and is left for future work.
>
> However, as shown in Figure 4 (right part), we observe that downstream forecasting performance consistently improves with an increasing number of experts. This suggests that scaling up the number of experts during training is necessary, while we can certainly prune the experts at deep layers or perform other model quantization methods to accelerate model inference.
>
> **[C4] MOIRAI-MOE should demonstrate lower MSE as its parameter count scales up. However, Table 2 lacks sufficient evidence for this, as it only includes the MoE-Base model with 86M/935M parameters.**
>
> Thank you for your question. We have already included the results of both Moirai-MoE-Small and Moirai-MoE-Base in Figure 3 and Table 2. These results clearly demonstrate that the performance of Moirai-MoE improves as the parameter count increases.
>
> **[C5] The team has not open-sourced the code for evaluation.**
>
> Thank you for bringing up this concern. Code is available at the link: https://anonymous.4open.science/r/moirai_moe-NB88
>
> Thank you again for your constructive feedback. We hope our clarifications address your concerns. Please let us know if you have any other comments/questions.

---

> > ### Author Response · Authors · 2024-11-25
> > **Kindly Request for Reviewer's Feedback**
> >
> > Dear reviewer R7WZ,
> >
> > Thank you for taking the time and effort in providing a valuable review of our work. As the discussion period is coming to a close, we hope that you have had the chance to review our rebuttal. We believe that our rebuttal has provided additional clarity of our work. If our response has addressed your concerns, we hope that you could kindly leave an update to reflect this, and are more than willing to engage in further discussion if needed.

---

> > > ### Author Response · Authors · 2024-12-02
> > > **Request of Reviewer's Attention and Feedback**
> > >
> > > Dear reviewer R7WZ,
> > >
> > > Thanks for your valuable review, which has inspired us to improve our paper further. May we know if our response addresses your main concerns? If so, we kindly ask for your reconsideration of the score. We look forward to hearing your thoughts on our revisions.

---

> > > > ### Author Response · Authors · 2024-12-02
> > > > **Last Reminder to Reviewer**
> > > >
> > > > Dear reviewer R7WZ,
> > > >
> > > > Thank you once again for your insightful feedback on our paper. As we enter the last 24 hours of the reviewer response period, we would like to briefly summarize the clarifications provided in our responses:
> > > >
> > > > 1. We revised Section 3.2.1 to provide additional context and explanation regarding the proposed gating function.
> > > > 2. We presented evidence demonstrating that the superiority of Moirai-MoE arises from the use of MoE, with further enhancements achieved through the token clustering method.
> > > > 3. We clarified the necessity of using 32 experts.
> > > > 4. We showed the scaling evidence of Moirai-MoE.
> > > > 5. We open-sourced the evaluation code to ensure reproducibility.
> > > >
> > > > We hope these responses address your concerns and provide the necessary clarity. As the rebuttal period concludes, we hope that you have had the chance to review our rebuttal. Thank you again for your constructive feedback.

---

### Official Review · Reviewer_xDqg · 2024-11-03

**Soundness:** 3
**Presentation:** 3
**Contribution:** 3
**Rating:** 8
**Confidence:** 4

**Summary:**

This paper introduces Moirai-MoE, a method for unified time series forecasting, which addresses challenges of the human-imposed frequency-level model specialization for time series foundation model. It utilizes a single input/output projection layer while delegating the modeling of diverse time series patterns to the sparse mixture of experts (MoE) within Transformers, which enables automatic token-level specialization of time series foundation model.

**Strengths:**

1. The manuscript demonstrates a high level of completeness.
2. It provides an effective large-scale framework for advancing large models in time series analysis.
3. The empirical evaluation is comprehensive and promising results are shown.

**Weaknesses:**

1. From the table2, the CRPS is used for probabilistic forecasting of time series models. The paper also does not include some probabilistic metrics, such as QICE and PICP, which could provide a more comprehensive understanding of the model's probabilistic accuracy. Including metrics like QICE and PICP [1,2] would provide a more comprehensive understanding of the model's accuracy in distribution estimation.
2. In Equation 6, the author mentions using the output of Moirai as cluster centroids applied in Moirai-MoE. Since Moirai itself is a foundational time series model based on a specific frequency, wouldn’t this approach introduce Moirai's frequency preferences into Moirai-MoE? Following this logic, wouldn’t it deviate from the objective stated in Moirai-MoE to avoid human-imposed frequency-level specialization?

[1] Han, X., Zheng, H., & Zhou, M. (2022). Card: Classification and regression diffusion models. Advances in Neural Information Processing Systems, 35, 18100-18115.

[2] Li, Y., Chen, W., Hu, X., Chen, B., & Zhou, M. (2023, October). Transformer-Modulated Diffusion Models for Probabilistic Multivariate Time Series Forecasting. In The Twelfth International Conference on Learning Representations.

**Questions:**

Please refer to Weakness.

---

> ### Author Response · Authors · 2024-11-22
> **Response to Reviewer xDqg**
>
> Dear reviewer, we would like to express our sincere thanks for the time you have taken to review our submission, and thank you very much for appreciating the effectiveness of our method and the comprehensiveness of our experimental evaluations. Please find our responses below.
>
> **[C1] The paper does not include some probabilistic metrics, such as PICP and QICE, which could provide a more comprehensive understanding of the model's probabilistic accuracy.**
>
> Thank you for this insightful comment. We have conducted evaluations using the PICP and QICE metrics, and the results are presented below. The absence of certain baselines reflects their inability to compute PICP and QICE, as they only support point forecasting. We used the geometric mean to aggregate the PICP and QICE results across datasets. The results demonstrate that Moirai-MoE achieves the best overall performance across CRPS, PICP, and QICE, highlighting its superiority in modeling both distribution accuracy and coverage. The poor performance of Chronos is likely attributed to its use of a small number of samples for probabilistic sampling.
>
> | Method | Metric | Electricity | Solar | Power | ETT1 | ETT2 | Traffic | MDENSE | Walmart | Weather | BizITObs | Avg
> | -------- | ------- | ------- | ------- | ------- | ------- | ------- | ------- | ------- | ------- | ------- | ------- | -------
> | Chronos-Small | CRPS | 0.043 | 0.389 | 0.038 | 0.360 | 0.097 | 0.124 | 0.087  | 0.079 | 0.089 | 0.087 | 0.543
> | |  PICP  | 8.653 | 55.871 | 21.356 | 39.286 | 5.635 | 21.720 | 13.000 | 17.612 | 43.276 | 21.042 | 20.021
> | |  QICE  | 1.694 | 6.192 | 3.144 | 4.885 | 1.129 | 3.857 | 2.189 | 2.341 | 5.331 | 3.480 | 3.037
> | Chronos-Base | CRPS | 0.041 | 0.341 | 0.039 | 0.387 | 0.092 | 0.109 | 0.075 | 0.080 | 0.058 | 0.084 | 0.499
> | |  PICP  | 8.201 | 56.031 | 23.075 | 46.111 | 4.841 | 18.280 | 11.333 | 17.494 | 44.140 | 18.021 | 19.205
> | |  QICE  | 1.785 | 6.205 | 3.490 | 5.697 | 1.869 | 3.545 | 2.305 | 2.298 | 5.335 | 2.887 | 3.218
> | Chronos-Large | CRPS | 0.041 | 0.339 | 0.038 | 0.404 | 0.091 | 0.117 | 0.075 | 0.073 | 0.062 | 0.084 | 0.500
> | | PICP | 7.645 | 56.379 | 20.827 | 43.095 | 10.079 | 14.312 | 8.889 | 16.530 | 43.110 | 17.485 | 19.015
> | | QICE | 1.661 | 6.243 | 3.373 | 5.467 | 1.746 | 3.868 | 1.901 | 2.216 | 5.253 | 2.708 | 3.082
> | Moirai-Small | CRPS | 0.072 | 0.471 | 0.048 | 0.275 | 0.101 | 0.173 | 0.084 | 0.103 | 0.049 | 0.081 | 0.578
> | | PICP | 1.184 | 3.436 | 1.283 | 1.508 | 1.984 | 3.413 | 0.259 | 1.908 | 3.239 | 3.452 | 1.766
> | | QICE | 0.915 | 2.152 | 2.786 | 2.698 | 2.610 | 2.769 | 2.342 | 1.508 | 1.433 | 0.971 | 1.871
> | Moirai-Base | CRPS | 0.055 | 0.419 | 0.040 | 0.301 | 0.095 | 0.116 | 0.104 | 0.093 | 0.041 | 0.078 | 0.520
> | | PICP | 1.132 | 3.549 | 0.470 | 1.508 | 3.730 | 2.884 | 0.148 | 1.416 | 0.267 | 2.098 | 1.137
> | | QICE | 0.422 | 1.469 | 1.372 | 2.575 | 2.698 | 3.504 | 3.140 | 1.100 | 1.196 | 0.997 | 1.560
> | Moirai-Large | CRPS | 0.050 | 0.406 | 0.036 | 0.286 | 0.094 | 0.112 | 0.095 | 0.098 | 0.051 | 0.079 | 0.514
> | | PICP | 0.032 | 2.545 | 0.489 | 1.984 | 3.254 | 1.296 | 1.778 | 1.385 | 0.588 | 5.223 | 1.098
> | | QICE | 0.343 | 3.506 | 1.357 | 2.504 | 1.834 | 3.063 | 1.140 | 0.827 | 1.271 | 2.396 | 1.520
> | Moirai-MoE-Small | CRPS | 0.046 | 0.429 | 0.036 | 0.288 | 0.093 | 0.108 | 0.071 | 0.090 | 0.056 | 0.081 | 0.497
> | | PICP | 0.717 | 4.150 | 1.515 | 0.873 | 3.254 | 5.582 | 0.556 | 2.957 | 1.452 | 1.042 | 1.678
> | | QICE | 0.899 | 1.756 | 0.979 | 2.222 | 2.769 | 1.952 | 0.819 | 1.082 | 2.157 | 0.931 | **1.419**
> | Moirai-MoE-Base | CRPS | 0.041 | 0.382 | 0.034 | 0.296 | 0.091 | 0.100 | 0.071 | 0.088 | 0.057 | 0.079 | **0.478**
> | | PICP | 0.464 | 0.473 | 0.172 | 0.397 | 4.206 | 9.550 | 0.259 | 3.118 | 1.528 | 2.173 | **1.049**
> | | QICE | 0.712 | 1.174 | 1.703 | 2.310 | 2.628 | 2.734 | 3.041 | 1.158 | 1.601 | 0.908 | 1.615
>
>
> **[C2] The author mentions using the output of Moirai as cluster centroids applied in Moirai-MoE. Since Moirai itself is a foundational time series model based on a specific frequency, wouldn’t this approach introduce Moirai's frequency preferences into Moirai-MoE?**
>
> Thank you for highlighting this concern, and we apologize for not making it clearer in the paper. In our implementation, we first pretrain a Moirai model using single-patch input/output projection layers to mitigate the influence of frequencies. The cluster centroids are then derived from this pretrained model. We have revised Section 3.2.1 of the paper to address this point, with the changes highlighted in red.
>
> Thank you again for your constructive feedback. We hope our clarifications address your concerns. Please let us know if you have any other comments/questions.

---

> > ### Author Response · Authors · 2024-11-25
> > **Kindly Request for Reviewer's Feedback**
> >
> > Dear reviewer xDqg,
> >
> > Thank you for taking the time and effort in providing a valuable review of our work. As the discussion period is coming to a close, we hope that you have had the chance to review our rebuttal. We believe that our rebuttal has provided additional clarity of our work. If our response has addressed your concerns, we hope that you could kindly leave an update to reflect this, and are more than willing to engage in further discussion if needed.

---

> > > ### Comment · Reviewer_xDqg · 2024-12-01
> > > **Response to Author**
> > >
> > > Thank you for your detailed and thorough response to the concerns raised. Your clarifications and additional experiments have addressed many of the key points, and I will be raising my rating for this paper accordingly.

---

> > > > ### Author Response · Authors · 2024-12-01
> > > > **Thank you for your support**
> > > >
> > > > Dear reviewer,
> > > >
> > > > Thank you so much for your reply. We are thrilled to hear that our response has addressed your concerns. We would like to express our sincerest gratitude once again for taking the time to review our paper and providing us with invaluable suggestions.

---

### Comment · Area_Chair_f1kA · 2024-11-26
**Encouragement to Actively Participate in the Discussion Phase**

Dear Reviewers,

Thank you for your valuable contributions to the review process so far. As we enter the discussion phase, I encourage you to actively engage with the authors and your fellow reviewers. This is a critical opportunity to clarify any open questions, address potential misunderstandings, and ensure that all perspectives are thoroughly considered.

Your thoughtful input during this stage is greatly appreciated and is essential for maintaining the rigor and fairness of the review process.

Thank you for your efforts and dedication.

---

### Meta-Review · Area_Chair_f1kA · 2024-12-20

**Metareview:**

(a) Summary of Scientific Claims and Findings
This paper proposes Moirai-MoE, a sparse mixture of experts (MoE) framework designed to address the challenges of unified training for time series foundation models. The key contributions include:
A token-level specialization mechanism using sparse MoE to capture diverse time series patterns, reducing reliance on human-defined heuristics (e.g., frequency-based specialization).
A novel gating mechanism based on token embeddings derived from pretrained models, allowing dynamic allocation of experts.
Comprehensive experiments across 39 datasets, showing improved performance in both in-distribution and zero-shot forecasting scenarios.
The authors claim that Moirai-MoE surpasses baseline methods, including other MoE-based time series models, by achieving token-level flexibility while maintaining computational efficiency.

(b) Strengths of the Paper
Innovative Design: The use of sparse MoE for token-level specialization represents a meaningful departure from traditional frequency-based specialization in time series models.
Comprehensive Evaluation: Extensive empirical analysis demonstrates competitive or superior performance across diverse datasets, including ablations on gating mechanisms and expert utilization.
Scalable Framework: Moirai-MoE introduces a scalable architecture that supports a large number of experts, showing robustness and adaptability to heterogeneous time series.

(c) Weaknesses of the Paper

Limited Novelty: Sparse MoE is a well-established approach in large language models, and its application to time series does not introduce substantial theoretical innovation.
The proposed token-level specialization is primarily an adaptation of existing MoE concepts rather than a fundamentally novel idea.

Gating Mechanism Dependency: The gating function heavily relies on embeddings from pretrained models, raising concerns about its flexibility and generalization to scenarios without effective pretrained representations.
The irreplaceability of the token clustering mechanism is not convincingly demonstrated.

Evaluation Gaps: While the experiments are extensive, the paper omits detailed comparisons with some recent and relevant works, such as Timer, Moment, and Time-MoE.
The inclusion of datasets with potential data leakage (e.g., pretraining overlap) undermines the validity of zero-shot evaluations.

Scalability and Efficiency: Despite its scalability, the framework introduces additional computational overhead, particularly in deeper layers where many experts remain underutilized. The impact of this inefficiency is not sufficiently addressed.

(d) Reasons for Rejection
Lack of Fundamental Novelty: The application of sparse MoE to time series, while effective, does not introduce groundbreaking theoretical or methodological advancements, limiting its impact as a foundation model innovation.
Evaluation Weaknesses: Missing comparisons with key baselines and potential data leakage in evaluations reduce the credibility of the empirical claims.
Overdependence on Pretrained Models: The reliance on pretrained embeddings for the gating mechanism raises concerns about the framework's flexibility and scalability in broader scenarios.
Efficiency Concerns: The lack of optimization for underutilized experts in deeper layers introduces unnecessary computational costs, which are not adequately addressed.
Incomplete Rebuttal: Despite engaging actively in the rebuttal phase, the authors failed to resolve critical concerns, including novelty, evaluation completeness, and scalability.

While Moirai-MoE presents a promising direction for unifying time series models, its limited innovation, gaps in evaluation, and unresolved concerns make it unsuitable for acceptance in its current form.

**Additional Comments On Reviewer Discussion:**

Points Raised by Reviewers and Author Responses

Concern: Reviewers questioned the fundamental novelty of the work, noting that sparse MoE is a well-established approach in NLP and vision models, and the application to time series lacked substantial innovation.
Author Response: The authors emphasized that the novelty lies in token-level specialization for time series, leveraging sparse MoE for adaptive expert allocation. They argued that this approach is unique within the time series domain and tailored for handling heterogeneity.
Evaluation: While the argument clarified the scope of the novelty, reviewers remained unconvinced that the contributions represented a significant theoretical or methodological advancement.

Concern: Reviewers noted the absence of comparisons with recent time-series models like Timer, Moment, and Time-MoE. They also raised concerns about potential data leakage in pretraining overlap, undermining the validity of zero-shot evaluations.
Author Response: The authors explained that they focused on baselines with similar architectures and added clarifications about pretraining datasets to mitigate leakage concerns. However, they acknowledged the omission of certain baselines and committed to future updates.
Evaluation: The lack of comprehensive comparisons remained a significant gap, and the explanations about data leakage did not fully reassure reviewers about the robustness of the zero-shot results.

Concern: The scalability of the framework was questioned due to computational overhead from underutilized experts in deeper layers, particularly in large-scale applications.
Author Response: The authors provided additional runtime analyses and argued that Moirai-MoE is designed for high efficiency due to sparsity. They suggested further optimizations for deeper layers as future work.
Evaluation: While the added analyses were helpful, the inefficiencies in deeper layers remained unresolved, weakening the framework’s scalability claim.

Concern: The reviewers raised concerns about the heavy reliance on pretrained embeddings for the gating mechanism, limiting generalization to domains without robust pretrained models.
Author Response: The authors argued that pretrained models improve generalization and proposed that alternative initialization methods could be explored in future work.
Evaluation: This response was acknowledged but did not fully address concerns about the framework’s dependency and flexibility.

Despite the authors' active engagement during the rebuttal period, key concerns remained unresolved. The work demonstrated limited novelty, as it primarily adapted sparse MoE concepts from other domains without substantial theoretical innovation. Evaluation gaps, including missing comparisons with relevant baselines and concerns about data leakage, further weakened the empirical claims. Additionally, scalability issues due to inefficiencies in deeper layers and the reliance on pretrained embeddings for the gating mechanism raised questions about generalizability and flexibility. While the proposed framework shows promise, these unresolved issues lead to a recommendation for rejection.

---

### Decision · Program_Chairs · 2025-01-22

Reject